# Relationship between gender roles, motherhood beliefs and mental health

**Maribel Delgado-Herrera**[☯], **Anabel Claudia Aceves-Gómez**[☯], **Azalea Reyes-Aguilar**[ID]*[☯]

Departamento de Psicobiología y Neurociencias, Laboratorio de Neurocognición Social, Facultad de Psicología, Universidad Nacional Autónoma de México, Circuito Ciudad Universitaria Avenida, Ciudad de México, México

☯ These authors contributed equally to this work.
* azalea@neurocogcialab.org

**Data Availability Statement:** All database files are available from the OSF database in the project "Relationship between gender roles, motherhood beliefs and mental health". You can find them at https://osf.io/md48n/.

## Abstract

Gender roles, as social constructs, play a significant role in shaping individuals' beliefs and attitudes, influencing various aspects of life, including perceptions and expectations surrounding motherhood. These beliefs, acquired through culture and society, can have an impact on our mental well-being. This research consists of three independent studies conducted in the Mexican population. In the first and second studies, we extended the Attitudes Towards Gender Roles Scale and Motherhood Beliefs Scale and performed psychometric validation through exploratory and confirmatory factor analysis. The aim of including additional items in both scales was to update these attitudes and beliefs in Mexican culture to avoid the traditionalist bias in both instruments. Finally, the third study examined the relationship between the new versions of both scales and symptoms of depression, anxiety, and Positive Psychological Functioning as indicators of mental health in women and men with and without children. Our findings revealed a significant association between higher levels of traditional attitudes towards gender roles and traditional motherhood beliefs, as well as between non-traditional attitudes towards gender roles and non-traditional beliefs about motherhood. Interestingly, we observed that traditional attitudes toward gender roles were associated with lower anxiety and depression scores, while non-traditional attitudes were associated with higher levels of depression. Furthermore, individuals who embraced non-traditional attitudes towards both gender roles and motherhood beliefs tended to exhibit better psychological well-being in all subsamples. Additionally, women generally showed lesser alignment with traditional attitudes towards both gender roles and motherhood beliefs compared to men. However, women reported higher rates of depression and anxiety, along with lower psychological well-being scores, than their male counterparts. This highlights the significant influence that traditional cultural norms about gender roles and motherhood have on women's mental health, underscoring the need for a deeper understanding and reevaluation of these traditional constructs in society.

**Funding:** This project was supported by UNAM-DGAPA-PAPIIT with the number: IN215521 for the corresponding author ARA. The funders had no role in study design, data collection and analysis, decision to publish, or preparation of the manuscript. https://dgapa.unam.mx/index.php/impulso-a-la-investigacion/papiit.

**Competing interests:** The authors have declared that no competing interests exist.

## Introduction

Gender roles, as social constructs, play a significant role in shaping people's beliefs and attitudes, influencing various aspects of life, including their perceptions and expectations, and even impacting their mental health and well-being.

In Western societies, gender ideology is molded by traditional gender roles defined by the sexual division of labor, where specific tasks are strictly assigned based on gender [1]. Women are typically tasked with reproductive work within the home and family, involving domestic management, infrastructure maintenance, and the care of family members. Conversely, men are expected to engage in productive work, contributing to the production of public goods and services, with economic retribution and social recognition [2].

This societal structure gives rise to traditional stereotypes of masculinity and femininity, representing the accepted beliefs about suitable characteristics for men and women [3]. Masculinity often involves traits like dominance, leadership, and competitiveness, while femininity is associated with qualities such as understanding, warmth, and compassion. These learned behaviors gradually become ingrained expectations, influencing individuals' actions and earning societal approval or disapproval [4], ultimately impacting their mental health and overall well-being.

In general, traditional masculinity traits have been associated with better physical health [5] and mental well-being. Specifically, traditional masculinity has shown a negative association with depression [6] and anxiety symptoms [7]. The influence of femininity traits on mental health remains inconclusive [8], as different studies have presented conflicting results. Some research suggests that femininity traits may increase the risk of depression in women [9], whereas other studies show a weak negative correlation between femininity and depression [6, 7]. In addition, there is evidence of a positive correlation between femininity traits and anxiety symptoms [7]. Notably, women experience higher rates of physical and mental health problems than men in different age groups and regions around the world [10].

Vespa [11] contends that gender ideology is dynamic, evolving over time as individuals encounter diverse social environments that introduce them to gender expectations related to aspects like marriage, parenthood, and work. Various determinants, such as race, age of children, education, family type, and pre-parenthood attitudes toward gender roles, contribute to this organic nature [11–13]. Fatherhood, for instance, often entails the expectation of being the primary breadwinner, whereas motherhood encompasses responsibilities related to childcare, nurturing, and housework [14]. Nevertheless, diverse experiences of parenthood influence gender ideology, shaped by the specific gender expectations individuals encounter. For married couples, parenthood tends to be associated with less egalitarian gender ideologies, irrespective of gender and race [11]. Conversely, some studies suggest that parenthood may reinforce traditional gender role stereotypes when compared to childless individuals [13]. Moreover, there is evidence indicating that individuals may develop more traditional attitudes and behaviors regarding gender roles once they become parents [15–17].

Attitudes toward traditional gender roles also impact individuals without children in different ways. For instance, women who prioritize their careers over starting a family may face judgments of being less feminine and more selfish, receiving more negative evaluations compared to women with children. Similarly, childfree men may encounter similar negative evaluations in comparison to men who have children [18]; or men who take on more domestic responsibilities may be seen as less manly [14]. These attitudes create obstacles for individuals who wish to adopt non-traditional gender roles. It has been reported that those who deviate from traditional gender roles may experience discrimination, leading to negative effects on both their mental and physical health [19].

Motherhood beliefs reflect societal norms and gender roles, yet studies often narrowly focus on the relationship between parenthood and mental health, neglecting the gendered aspects of motherhood. In other words, they often overlook the impact of gender roles and social expectations on this experience. For women, motherhood carries a unique context in which it is seen as both a gender-based obligation and a defining aspect of identity [20, 21]. Moreover, motherhood, unlike fatherhood, is often described as intensive, meaning that it is seen as the most significant and valuable role a woman can assume. At the same time, it imposes strict demands and sets unattainable standards, such as the invisible double workload involving both productive and reproductive work [2, 22].

The imposition of these roles, whether one is a mother or not, poses challenges for those diverging from societal expectations. Warner [23] introduces the "mommy mystique," illustrating how mothers grapple with stress, anger, and guilt due to unrealistic motherhood expectations. Intensive motherhood ideology has been linked to lower self-efficacy, high stress levels [24], parental exhaustion [25], depression, reduced life satisfaction, low perceived family support [26], and anxiety [27].

In Mexico, family and gender roles adhere strictly to societal expectations, imposing significant pressure to conform. A central cultural aspect is affiliative obedience, where both men and women embody the distinct feature of Mexican culture: self-sacrifice. This implies a belief that prioritizing the needs of others over one's own is more important [28]. Mexicans frequently prioritize external social demands over their personal interests and desires [29].

The traditional Mexican family serves as an illustrative context to understand how gender roles are instilled and acquired in Mexico. According to Diaz-Loving et al. [28], Mexican families are guided by two main axes: a) the power and absolute supremacy of the father, and b) the love and essential sacrifice of the mother. This family structure is mirrored in normative traits of Mexicans, including the emphasis on obedience, the fear of family dishonor, and the magnification of virginity even among adolescents. Moreover, predominant beliefs in this culture include machismo, fear of parental authority, and the importance of respect.

Nevertheless, in Mexico there is a growing diversification of family structures, contemporary meanings related to the family are going beyond the exclusivity of the nuclear biparental model and there is greater openness and tolerance regarding possible alternative forms of coexistence and personal and family development [30]. Within this diversity, coupled with the collectivist character of Mexican society [31], is the participation of networks of women and extended family in the care of children, a term known as *othermothers* [32]. This could facilitate some conditions such as women having a support network that allows them to occupy other professional and work roles or to have time for self-care, reducing the parental burden.

Despite current changes in parenting roles and parenting decisions, it seems that there is an underlying maintenance of traditional gender roles, and if so, it would be important to know their relationship with mental health in both people with and without children. Furthermore, there is a lack of research on what this relationship would be like in a society where traditional family and gender beliefs are highly esteemed, and where women, due to the conventional family structure, are engaged in childcare collectively rather than individually. To test this hypothesis, the first step would be to have updated instruments that probe attitudes towards gender roles today, otherwise the results between gender roles and mental health could be biased.

The present research was divided into three independent studies. In the first and second studies, we extended the Attitudes Towards Gender Roles Scale (ATGRS) [33] and Motherhood Beliefs Scale (MBS) [34] and performed psychometric validation through exploratory and confirmatory factor analysis. The aim of including additional items in both scales was to update attitudes and beliefs in Mexican culture in order to avoid the traditionalist bias in both

instruments. Finally, the third study examined the relationship between the new versions of ATGRS and MBS with measures assessing depression, anxiety and Positive Psychological Functioning (PPF) as indicators of mental health. This analysis was conducted in women and men with and without children.

## Study 1

The objective for this first study was to add and validate new items of the ATGRS [33].

### Participants

Participants were recruited through the laboratory's social media platforms (Instagram, Facebook, Twitter) and the personal social media accounts of the lab members from November to December 2021. Participants were native-Spanish speakers of Mexican nationality and 18 years or older and signed a written privacy notice and an informed consent form. During the data collection, demographic variables such as age, sex, parental status, and level of education were recorded in an anonymized manner. Later, during the database construction, one of the authors ensured that the data was anonymized by removing any information that could identify the participants, in order to perform the analysis and interpretation of the results. The research protocol received approval from the local Ethics in Research Committee of the School of Psychology-UNAM, in accordance with the federal guidelines set forth by the Mexican Health Department, which align with international regulations.

### Instruments and procedure

We selected the ATGRS for our study as it is the sole scale developed in Mexico specifically designed to evaluate the construct it measures: an individual's perception based on societal norms and expectations related to traditional and non-traditional gender roles in men and women [33]. The authors conducted an exploratory study prior to developing the scale to ensure its content was both culturally sensitive and had face validity. They used open questions to collect diverse conceptions -encompassing behaviors, traits, beliefs, etc.- that men and women have regarding their gender identity [33]. The scale's validation process involved 120 men and 224 women, with an average age of 30 years (SD = 9.91). The educational background of the participants was varied: 40% had an education level of high school or less, while 60% had attained at least a bachelor's degree. All participants belonged to a medium socioeconomic stratum.

ATGRS comprises 21 items rated on a 5-point Likert scale (1 = I like it very much, 5 = I dislike it very much). The scale is structured around three factors, collectively accounting for 46% of the variance. The first factor, Traditional Attitudes Towards Gender Roles, includes statements that endorse the continuation of conventional gender roles. The second factor, Favorable Attitudes Towards Gender Equity, reflects a supportive view of equal rights and opportunities for both men and women. The third factor, Favorable Attitudes Towards Female Empowerment, consists of items that positively assess the progress and emancipation of women. The Cronbach's alpha coefficients, which measure internal consistency, were high, indicating strong reliability: 0.86 for the first factor, 0.85 for the second, and 0.76 for the third.

Our analysis identified several issues with the composition of the factors within the ATGRS. Firstly, the scale includes items that are anchored in hegemonic gender roles. Even the second factor, which is intended to assess favorable attitudes towards gender equity, does so within a traditional framework. For instance, the item "That the man participates in the care of the children" reflects the conventional view that caregiving is primarily a woman's role. Thus, within the scale, male involvement in childcare is seen as an equitable attitude. However, in a modern context, true equity in childcare would mean recognizing both parents' rights and

responsibilities in child-rearing. This suggests a need for additional items in this factor to more accurately encapsulate the essence of gender equity.

Another issue is with the third factor, Favorable Attitude Towards Female Empowerment. Most of its items seem to focus more on gender equity rather than on female empowerment. Take, for example, "That women have the same freedom as men" (Item 3 in the original scale) and "That men and women perform the same tasks" (Item 6 in the original scale). Considering empowerment as a process enabling women, particularly those who have faced oppression, to make independent and strategic life choices based on their personal priorities [35], it becomes evident that the items in this third factor fall short of adequately capturing the concept of female empowerment.

To address these issues, we modified the items that make up factor 1 of the original scale, Favorable attitude towards traditional roles, but with reversed roles. For instance, the original item "That the man establishes the rules of the home" was complemented with "That the woman establishes the rules of the home." In the second factor, Favorable Attitude Towards Gender Equity, we introduced role-reversed items such as "That the woman expresses her emotions just like the man" and "That the man focuses on personal and professional self-improvement." Additionally, we incorporated two items related to maternity/paternity to address specific gender role stereotypes associated with women [21]. These items were "That the success of being a woman lies in being a mother" and "That the success of being a man lies in being a father." This inclusion acknowledges the gendered expectations surrounding parenting roles. In total, 13 new items were added to the scale, bringing the total number of items in the extended version of the ATGRS to 34. All participants in our study responded to this expanded scale via a Google Form.

## Data analysis

To assess the adequacy of the sample, we performed a Kaiser-Meyer-Olkin (KMO) test and Bartlett's Test of Sphericity.

To achieve the objective of this initial study, which is to psychometrically validate the ATGRS with new items, we conducted an exploratory factor analysis followed by a confirmatory factor analysis as outlined below. We employed two approaches to determine the appropriate number of factors: Kaiser's criteria and a scree plot. According to Kaiser's criteria, we selected factors with eigenvalues greater than 1.0. In the scree plot, we identified the point where the eigenvalues seemed to stabilize, referred to as the "elbow" of the graph. This indicated the number of factors to include in the exploratory analysis. After considering the results from both approaches and taking theoretical foundations into account, we determined the final number of factors for the exploratory factor analysis.

For factor extraction, we used an orthogonal rotation since the factors were theoretically unrelated. Subsequently, we examined the factor loadings, retaining items with coefficients greater than 0.40 and eliminating those with lower loadings.

Next, we conducted a Confirmatory Factor Analysis with a Satorra-Bentler scaled correction of maximum likelihood to validate the factors identified in the exploratory analysis. Finally, we assessed the model using Chi-square goodness-of-fit statistics ($p > .05$), the Comparative Fit Index (CFI), the Tucker-Lewis Index (TLI), the Root Mean Squared Error of Approximation (RMSEA), and the Standardized Root Mean Square Residual (SRMR). All statistical analyses were performed using R software, version 4.1.1 [36].

## Results

The sample for Study 1 consisted of 2,677 participants, including 1,742 females (65.07%), 927 males (34.62%), and eight non-binary individuals (0.29%). The participants had a mean age of

23.36 years with a standard deviation of 7.76, and with an age range of 18–50 years. In terms of educational attainment, the distribution in the sample was as follows: seven participants (0.26%) with a primary school education, 79 (2.95%) with a secondary school education, 1,560 (58.27%) with a high school education, 952 (35.56%) with a bachelor's degree, and 79 (2.95%) with a postgraduate degree. Among the participants, 7.39% reported having more than one child, 5.19% reported having only one child, and 87.41% reported not having any children.

## Exploratory factor analysis

The KMO test was 0.89, and Bartlett's Test of Sphericity was significant ($\chi2$ = 68271.12; df = 561, p < .001), both results suggesting that the data was adequate for performing exploratory factor analysis. The scale´s reliability with the new items added was Cronbach's alpha = 0.91.

Kaiser's criteria revealed the presence of seven factors. The scree plot suggested seven factors first, followed by four factors. Based on these approaches, we decided to extract four factors because it was the model that best fitted to the theoretical bases of the topic.

Using the four-factor structure, we conducted an orthogonal rotation and eliminated items based on established factor loading criteria. Three items were removed, resulting in a final scale of 31 items across four factors (Table 1). The reliability of the final scale, consisting of 31 items, remained high with a Cronbach's alpha of 0.91.

The four factors were composed as follows, presented in order from the factor encompassing the most traditional attitudes to the factor that includes non-traditional attitudes: Factor 1, *Traditional attitudes towards gender roles*, (eight items); Factor 2, *Favorable attitude towards egalitarian success*, (four items); Factor 3, *Favorable attitude towards gender equity*, (12 items); and, Factor 4, *Non-traditional attitudes towards gender roles* (seven items) (Table 1).

## Confirmatory factor analysis

The fit indices calculated to evaluate the model indicated that the four-factor structure derived from the exploratory analysis was validated. The obtained results were as follows: CFI = 0.77, TLI = 0.75, RMSEA = 0.11, SRMR = 0.07, and S-B$\chi2$ (428) = 14464.42 (p < 0.001). Cronbach's alpha coefficients were 0.86 for Factor 1, *Traditional attitudes towards gender roles*; 0.86 for Factor 2, *Favorable attitude towards egalitarian success*; 0.94 for Factor 3, *Favorable attitude towards gender equity*; and 0.85 for Factor 4, *Non-traditional attitudes towards gender roles*. The factor loadings of each item are shown in Table 1.

## Discussion of study 1

In this first study, we identified two issues with the ATGRS scale, prompting us to propose an extension. The first issue pertains to factor 2 of the original scale, where we observed a bias towards traditional content in the items. The second issue concerns the third factor of the original scale, labeled as female empowerment, as the items reflected behaviors associated more with gender equity rather than female empowerment.

Through our exploratory and confirmatory analyses, incorporating the newly proposed items, we found that the scale required a restructuring into four factors. This restructuring successfully addressed the aforementioned issues.

In the ATGRS extended scale, all the items of factor 1 of the original scale were retained, now referred to as *Traditional attitudes towards gender roles*.

In factor 2, we grouped four items that address the success of men and women, and we named this factor *Favorable attitudes towards egalitarian success*. An item from factor 1 of the original scale, (The success of being a man lies in having a paid job*)*, was included in this

**Table 1. Items from original and extended ATGRS.**

| Item number on original scale | Factors | New/ original item | OriginalFactor | Loadings Std | |
|---|---|---|---|---|---|
| | | | | Exploratory | Confirmatory |
| | **Factor 1. Traditional attitudes towards gender roles** | | | | |
| 1 | The man establishes the rules of the home / *Que el hombre establezca las reglas del hogar* | Original | 1 | 0.65 | 0.63 |
| 4 | The man always has the last Word / *Que el hombre tenga siempre la última palabra* | Original | 1 | 0.63 | 0.62 |
| 7 | The woman is in charge of cooking / *Que sea la mujer la que se encargue de hacer la comida* | Original | 1 | 0.63 | 0.6 |
| 10 | The man is dominant / *Que el hombre sea dominante* | Original | 1 | 0.55 | 0.51 |
| 13 | The woman is in charge of taking care of the children / *Que la mujer se encargue del cuidado de los hijos(as)* | Original | 1 | 0.54 | 0.53 |
| 15 | The woman dedicates herself to housework and stays at home / *Que la mujer se dedique a las tareas domésticas y permanezca en el hogar* | Original | 1 | 0.57 | 0.65 |
| 17 | The man is a strong part of the relationship / *Que el hombre sea la parte fuerte de la relación* | Original | 1 | 0.57 | 0.59 |
| 19 | The woman is submissive and selfless / *Que la mujer sea sumisa y abnegada* | Original | 1 | 0.42 | 0.48 |
| | **Factor 2. Favorable attitude towards egalitarian success** | | | | |
| 21 | The success of being a man lies in having a paid job / *Que el éxito de ser hombre radique en tener un trabajo remunerado* | Original | 1 | 0.51 | 0.85 |
| NA | The success of being a woman lies in having a paid job / *Que el éxito de ser mujer radique en tener un trabajo remunerado* | New | NA | 0.50 | 0.89 |
| NA | The success of being a woman lies in being a mother / *Que el éxito de ser mujer radique en ser madre* | New | NA | 0.89 | 0.46 |
| NA | The success of being a man lies in being a father / *Que el éxito de ser hombre radique en ser padre* | New | NA | 0.86 | 0.46 |
| | **Factor 3. Favorable attitude towards gender equity** | | | | |
| 2 | The man participates in childcare / *Que el hombre participe en el cuidado de los hijos* | Original | 2 | 0.78 | 0.65 |
| 5 | The woman has job opportunities similar to men / *Que la mujer tenga oportunidades laborales similares a los hombres* | Original | 2 | 0.85 | 0.77 |
| 8 | The man expresses his emotions just like a woman / *Que el hombre externe sus emociones igual que una mujer* | Original | 2 | 0.81 | 0.71 |
| NA | The woman expresses her emotions just like a man / *Que la mujer externe sus emociones igual que un hombre* | New | NA | 0.68 | 0.52 |
| 11 | The woman overcomes herself personally and professionally / *Que la mujer se supere personal y profesionalmente* | Original | 2 | 0.94 | 0.88 |
| NA | The man overcomes himself personally and professionally / *Que el hombre se supere personal y profesionalmente* | New | NA | 0.94 | 0.85 |
| 14 | The man interacts and plays with the children / *Que el hombre conviva y juegue con los hijos* | Original | 2 | 0.92 | 0.85 |
| 16 | The woman develops outside the home environment / *Que la mujer se desarrolle fuera del ámbito hogareño* | Original | 2 | 0.86 | 0.79 |
| 3 | The woman has the same freedom as man / *Que la mujer tenga igual libertad que el hombre* | Original | 3 | 0.89 | 0.85 |
| 6 | Men and women perform the same tasks / *Que los hombres y las mujeres desarrollen las mismas tareas* | Original | 3 | 0.85 | 0.78 |
| 9 | The woman is self-sufficient / *Que la mujer sea autosuficiente* | Original | 3 | 0.90 | 0.85 |
| 12 | The woman participate in decision-making / *Que la mujer participe en la toma de decisiones* | Original | 3 | 0.90 | 0.85 |
| | **Factor 4, Non-traditional attitudes towards gender roles** | | | | |
| NA | The woman establishes the rules of the home / *Que la mujer establezca las reglas del hogar* | New | NA | 0.67 | 0.66 |

*(Continued)*

**Table 1.** (Continued)

| Item number on original scale | Factors | New/ original item | OriginalFactor | Loadings Std | |
|---|---|---|---|---|---|
| | **Factor 1. Traditional attitudes towards gender roles** | | | Exploratory | Confirmatory |
| NA | The woman always has the last word / *Que la mujer tenga siempre la última palabra* | New | NA | 0.65 | 0.65 |
| NA | The man is in charge of cooking / *Que sea el hombre el que se encargue de hacer la comida* | New | NA | 0.65 | 0.67 |
| NA | The woman is dominant / *Que la mujer sea dominante* | New | NA | 0.66 | 0.60 |
| 18 | The man is in charge of taking care of the children / *Que el hombre se encargue del cuidado de los hijos(as)* | Original | 2 | 0.65 | 0.67 |
| NA | The man dedicates himself to housework and stays at home / *Que el hombre se dedique a las tareas domésticas y permanezca en el hogar* | New | NA | 0.67 | 0.63 |
| NA | The woman is a strong part of the relationship / *Que la mujer sea la parte fuerte de la relación* | New | NA | 0.66 | 0.63 |
| | **Delated items** | | | | |
| NA | The man is submissive and selfless / *Que el hombre sea sumiso y abnegado* | New | NA | > 0.40 | |
| 20 | The man is more time away from home / *Que el hombre se encuentre más tiempo fuera del hogar* | Original | 1 | > 0.40 | |
| NA | The woman spends more time away from home / *Que la mujer se encuentre más tiempo fuera del hogar* | New | NA | > 0.40 | |

NA, Not Applicable

factor. Additionally, we grouped its counterpart, the idea that the success of women is having a paid job (a new item proposed by us).

Furthermore, we grouped two new items in factor 2. Continuing with our intention to include non-hegemonic statements, considering that the idea of success for women has traditionally focused on their role as mothers, we included an item in this factor that reflects the success of men in terms of their paternity: The success of being a man lies in being a father. As a result, factor 2 of the extended version of ATGRS incorporates two traditional items regarding the success of being a man or a woman, while also highlighting the non-hegemonic version of success in women and men.

In the extended version, factor 3 was made up of six items related to gender equity, which were taken from factor 2 of the original scale. As anticipated, the items that originally comprised factor 3, referred to as female empowerment, were grouped into the new factor 3, which we named *Favorable attitude towards gender equity*. Additionally, two new items were added (The woman expresses her emotions just like a man; and, The man overcomes himself personally and professionally). The extended version of factor 3 comprised a total of 12 items, surpassing the original scale's seven items, thus providing a more comprehensive representation of gender equity.

Lastly, all the items we added as counterparts to the traditional items were grouped in factor 4, named *Non-traditional attitudes towards gender roles*. It is important to mention that in this factor an item from factor 2 of the original scale (The man is in charge of taking care of the children).

In summary, the extended version of ATGRS consists of four factors, ranging from the most traditional gender attitudes (Factor 1) to the least traditional attitudes (Factor 4). Furthermore, non-hegemonic items were incorporated to assess the success of women and men (Factor 2), and the number of items addressing gender equity (Factor 3) was expanded.

The importance of expanding this scale lies in two main reasons. The first reason is to broaden the traditional conception of success for men and women. By acknowledging that a

woman's success can lie in having a paid job, it helps to challenge the stereotype that her role should be limited to child-rearing or being a housewife [37]. It also aims to eliminate the notion that a woman is only considered successful once she becomes a mother. This extended version will allow for the study of a realistic portrayal of the acceptance of women's lives today and the actual adoption of non-traditional gender roles. This can help to recognize and address any lingering biases towards what is considered feminine or masculine, with the aim of reducing them. Secondly, by exploring attitudes towards gender roles in a society where traditional gender roles are being challenged by the demands of social reorganization, we can gain insights into how this transformation of roles impacts mental health and overall quality of life for individuals.

## Study 2

The objective for this study was to create and validate new items of the MBS [34].

### Participants

We recruited a new sample of participants than in Study 1, from January to June 2022. In the same way, volunteers were Mexican, native-Spanish speakers and 18 years or older. All volunteers signed a written privacy notice and an informed consent form according to the approval of the same protocol as study 1, by the local Ethics in Research Committee of the School of Psychology-UNAM. Again, demographic variables were collected anonymously during data collection, and one author ensured this during database formation.

### Instruments and procedure

In Mexico there is just one validated scale that evaluates Motherhood Beliefs, i.e. MBS [34]. This scale was developed through interviews with three distinct groups of women: 1) those receiving gynecological care at a reproductive health hospital, 2) mothers of preschool children, and 3) university students. These women were asked about their perceptions of what it means to be a mother. Based on their responses, 17 items were written with Likert-type responses ranging from 0 = disagree to 5 = totally agree belonging to two dimensions: 1) Sense of Life, which assesses statements related to motherhood as a woman's life purpose, something that brings her happy and fulfilled; and 2) Social Duty, which includes statements about motherhood as an obligation to society, with a woman's value being diminished if she does not fulfill this duty. After initial development, three judges assessed how well each item aligned with the definitions of these proposed dimensions, leading to the elimination of two items.

The scale's validation involved a sample of 545 women, aged between 17 and 52 years (Mean = 30.97; SD = 7.54). To establish concurrent validity, the study included women with varying maternal statuses: those who experienced perinatal loss (35.8%), those undergoing infertility treatments or with high-risk pregnancies (10.8%), mothers of preschool-age children (36.6%), and female university students without children (17.1%). Educational background varied, with 64.7% having at least a secondary school education. Only 38.2% were employed, and about 40% were single. Approximately 40% of the participants had between one and five children.

Two additional items were excluded due to their factor loadings being less than 0.40 in the exploratory analysis, resulting in the final scale comprising 13 items. Again, the scale was categorized into two factors: 1) Sense of Life, and 2) Social Duty. The scale demonstrated satisfactory internal consistency, with a Cronbach's alpha of 0.92 for the total scale, 0.91 for the Sense of Life factor, and 0.83 for the Social Duty factor.

During the validation of the Motherhood Belief Scale, groups were contrasted based on maternal status. Findings revealed that women who had experienced perinatal loss or infertility issues exhibited more favorable beliefs towards motherhood on both subscales than did mothers and university students.

Although the initial validation included women with diverse motherhood statuses, we consider that the 13 items across just two factors are insufficient for a comprehensive assessment of motherhood in our society [37]. The original scale's items also displayed a significant bias towards hegemonic gender stereotypes, exemplified by statements like "A woman is complete until she is a mother".

To enhance the scale, we introduced additional items that more thoroughly examine various aspects of motherhood while avoiding gender stereotypes. These new items were derived from an extensive review of theoretical and scientific literature on motherhood from 2000 to 2021. This review led to the identification of a wide range of definitions, concepts, and beliefs about motherhood, which were then adapted into Likert-type statements. As a result, 16 new items were added to the MBS, covering areas such as female identity, decision-making, lifestyle, success, life plans, and emotions including love, frustration, pain, and passion. For example, new items included 'Maternity is a woman's decision' and 'Motherhood can generate anxiety,' as detailed in Table 2. Participants completed this extended version of the MBS via a Google Form.

## Data analysis

To attain the objective of psychometrically validating the MBS with new items, we followed the same analysis as Study 1, which involved an exploratory factor analysis followed by a confirmatory factor analysis as outlined below. For the statistical analysis of the psychometric properties of the extended MBS, we utilized R software, version 4.1.1 [29]. The analysis included several measures: the Kaiser-Meyer-Olkin (KMO) test, which assesses the suitability of the data for factor analysis; Bartlett's sphericity test, which examines the presence of redundancy or correlation among item scores in the extended MBS that can be summarized with a few factors; and Cronbach's α, which measures the reliability of the scale. Subsequently, we conducted an exploratory factor analysis to examine the structure of the extended MBS. Finally, we performed a confirmatory factor analysis (CFA) with a Satorra-Bentler scaled correction of maximum likelihood to assess the factor model's fit, as it provides a more robust fit [38] for extended-MBS.

## Results

The sample for Study 2 consisted of 565 participants, with 417 (73%) females, 144 (25.5%) males, and 4 (0.71%) identifying as non-binary. The age range of the participants was 18 to 50 years old (mean = 25.50, sd = 9.58). Among the participants, 19.3% (109) had children, while 80.7% (456) did not. The educational distribution of the sample was as follows: one participant (0.17%) with a primary school education, 7 (1.24%) with a secondary school education, 287 (50.8%) with a high school education, 251 (44.4%) with a bachelor's degree, and 19 (3.36%) with a postgraduate degree.

## Exploratory factor analysis

The exploratory factor analysis for psychometric properties of the extended-MBS showed that the adequacy of the sample was assessed as KMO = 0.86. The Bartlett's Test of Sphericity yielded a significant result ($\chi2 = 7500.40$; df = 406, p < 0.001), indicating that there was

**Table 2. Items from original and extended MBS.**

| Item number on original scale | Factors | New/ original item | Original Factor | Loadings Std | |
|---|---|---|---|---|---|
| | | | | Exploratory | Confirmatory |
| | **Factor 1: Social duty / Original-MBS** | | | | |
| 1 | A woman's value depends on her being a mother. / *El valor de una mujer depende de que sea madre.* | Original | 2 | 0.76 | 0.79 |
| 13 | If a woman does not have children, she deserves to be rejected by others. / *Si una mujer no tiene hijos, merece el rechazo de los demás.* | Original | 2 | 0.67 | 0.70 |
| 15 | For a woman life is only worthwhile if she has children. / *Para una mujer vale la pena vivir solo si tiene hijos.* | Original | 2 | 0.64 | 0.71 |
| 6 | A woman's duty is to have children. / *El deber de una mujer es tener hijos.* | Original | 2 | 0.45 | 0.58 |
| 8 | Men respect a woman more when she is a mother. / *Los hombres respetan más a la mujer cuando es madre.* | Original | 2 | 0.10 | Delated item |
| | **Factor 2: Sense of life / Original-MBS** | | | | |
| 9 | In order to feel happy a woman needs to have a child. / *Para sentirse feliz, una mujer necesita tener un hijo.* | Original | 1 | 0.80 | 0.88 |
| 5 | A woman isn't complete/fulfilled until she becomes a mother. / *Una mujer está completa hasta que es madre.* | Original | 1 | 0.80 | 0.87 |
| 4 | For a woman, no achievement compares to being a mother. / *Para una mujer, ningún logro se compara con ser madre.* | Original | 1 | 0.78 | 0.77 |
| 12 | The most important thing for a woman is to become a mother. / *Lo más importante para una mujer es ser madre.* | Original | 1 | 0.78 | 0.75 |
| 3 | A woman's greatest desire is to have one or more children. / *Lo que más desea una mujer es tener uno o más hijos.* | Original | 1 | 0.77 | 0.76 |
| 7 | A woman is happier if she becomes a mother. / *Una mujer es más feliz si es madre.* | Original | 1 | 0.77 | 0.75 |
| 11 | A woman doesn't feel fulfilled until she has a child. / *Una mujer se realiza hasta que tiene un hijo.* | Original | 1 | 0.76 | 0.86 |
| 2 | Life is worth living if you have children. / *La vida vale la pena si tienes hijos.* | Original | 1 | 0.55 | 0.57 |
| | **Factor 3: Hegemonic stereotypes / Extended-MBS** | | | | |
| NA | Motherhood makes women more responsible. / *La maternidad hace más responsable a la mujer.* | New | NA | 0.43 | 0.64 |
| NA | Motherhood bestows happiness. / *La maternidad otorga la felicidad.* | New | NA | 0.38 | 0.55 |
| NA | It is better when only the mother takes care of the children's upbringing. / *Es mejor que solo la madre se haga cargo de la crianza de los hijos(as).* | New | NA | 0.37 | 0.37 |
| NA | Motherhood implies forgetting the needs of women. / *La maternidad implica olvidar las necesidades propias de la mujer.* | New | NA | 0.35 | 0.22 |
| NA | Maternity interferes with the work-professional performance of women due to the pregnancy and parenting time. / *La maternidad interfiere con el desempeño laboral-profesional de las mujeres debido al tiempo de gestación y de crianza.* | New | NA | 0.32 | 0.20 |
| NA | Prioritizing your own needs before those of your children makes you a bad mother. / *Priorizar las necesidades propias antes que las de los hijos (as) te hace ser una mala madre.* | New | NA | 0.31 | 0.36 |
| | **Factor 4: Negative Emotions / Extended-MBS** | | | | |
| NA | Motherhood can be scary. / *La maternidad puede generar miedo.* | New | NA | 0.87 | 1.02 |
| NA | Motherhood can generate anxiety. / *La maternidad puede generar ansiedad.* | New | NA | 0.85 | 0.79 |
| | **Factor 5: Motherhood as a decision / Extended-MBS** | | | | |
| NA | A woman can have a successful identity without being a mother. / *Una mujer puede tener una identidad exitosa sin ser madre.* | New | NA | 0.69 | 0.73 |
| NA | Motherhood is one way to build a family, but it is not the only one. / *La maternidad es solo una forma de construir una familia, pero no la única.* | New | NA | 0.66 | 0.59 |
| NA | A couple's relationship can be solid or consolidated without children. / *Una relación de pareja puede ser sólida o consolidarse sin hijos.* | New | NA | 0.63 | 0.69 |

*(Continued)*

**Table 2.** (Continued)

| Item number on original scale | Factors | New/ original item | Original Factor | Loadings Std | |
|---|---|---|---|---|---|
| | Factor 1: Social duty / Original-MBS | | | Exploratory | Confirmatory |
| NA | Feminine identity is NOT built from the women's reproductive capacity. / *La identidad femenina NO se construye a partir de la capacidad reproductiva de las mujeres.* | New | NA | 0.62 | 0.61 |
| NA | A woman's identity is defined by personal, family, social, and professional diversity and not only from motherhood. / *La identidad de una mujer se define desde una diversidad personal, familiar, social, profesional y no sólo desde la maternidad.* | New | NA | 0.59 | 0.70 |
| NA | Motherhood is a woman's decision. / *La maternidad es una decisión de la mujer.* | New | NA | 0.57 | 0.55 |
| NA | Motherhood does not define a women's life. / *La maternidad no define la vida femenina.* | New | NA | 0.56 | 0.50 |
| NA | Other woman can raise the child besides the mother. / *La crianza puede ser ejercida por otras mujeres además de la mamá.* | New | NA | 0.35 | 0.36 |

NA, Not Applicable

significant correlation among the item scores. The reliability of the new scale, as measured by Cronbach's alpha coefficients, was 0.82.

To determine the number of factors, we applied Kaiser's criteria (eigenvalues > 1), which indicated the presence of five factors. Subsequently, employing an orthogonal rotation with the five-factor structure and minimizing the overall residual matrix through an ordinary least squares (OLS) procedure with empirical first derivatives, we confirmed the exclusion of the item with the lowest loading score from the original-MBS, i.e., item 13: *"Men respect a woman more when she is a mother."* The final scale, consisting of 28 items, demonstrated good reliability with a Cronbach's alpha coefficient of 0.87.

## Confirmatory factor analysis

The confirmatory factor analysis of extended-MBS demonstrated that the five-factor model exhibited a good fit according to the standard fit indexes, with a CFI = 0.84, RMSEA = 0.08, SRMR = 0.07, and S-Bχ2 (340) = 1499.32 (p < 0.001).

After confirming that the 28 items were allocated into five factors, we named and ordered them based on the degree of traditionalism, from most to least traditional: Factor 1, *Social duty*, included four items from original-MBS; Factor 2, *Sense of life*, comprised eight items from original-MBS; Factor 3, *Hegemonic stereotypes* comprised six new items; Factor 4, *Negative emotions*, included two new items; and, Factor 5, *Motherhood as a decision*, encompassed eight new items (Table 2).

Cronbach's alpha coefficients were 0.89 for *Social duty*, 0.91 for *Sense of life*, 0.54 for H*egemonic stereotypes*, 0.77 for *Negative emotions*, and 0.78 for *Motherhood as a decision*. The factor loadings for each item are presented in Table 2.

## Discussion of study 2

In our study, we aimed to refine the Motherhood Belief Scale (MBS) due to its original version's bias, which was narrowly concentrated on two factors: *Sense of Life* and *Social Duty*. These factors predominantly echoed conventional stereotypes about motherhood. To rectify this, the updated MBS preserved these two original factors while significantly broadening its framework. A thorough factorial analysis led to the integration of all 28 items, encompassing both the original and newly added items. We then named the factors in alignment with the

themes of the grouped items, thus including *Hegemonic stereotypes*, *Negative emotions*, and *Motherhood as a decision* alongside the two original and initial factors, *Sense of life* and *Social duty*. This expanded and diversified structure of the MBS offers a deeper, more inclusive exploration of motherhood's complex and varied dimensions.

The third factor, named *Hegemonic Stereotypes*, aims to underscore the predominant beliefs about motherhood that reinforce the exclusive responsibility of mothers in child-rearing. It reflects societal expectations of full-time dedication as a criterion for being deemed a good mother, emphasizing complete responsibility and happiness in motherhood, absolute commitment to procreation, and the act of relinquishing one's own needs and the professional field. Although *Hegemonic Stereotypes* are expressed across these diverse contexts (responsibility, parenting style, professional work, personal needs). The inclusion of various contexts in the items of this factor may have contributed to its lower consistency. Future efforts could focus on refining this factor by either narrowing down the included contexts or incorporating more items for each context.

However, the primary contribution of the extended MBS version was the inclusion of items representing the non-hegemonic perspective of motherhood, grouped in the last two factors. Factor Four, named *Negative Emotions*, encompassed items acknowledging the presence of negative emotions in motherhood, contrary to traditional approach that took for granted that motherhood to be only positive emotions. This conceptualization had not been previously addressed in motherhood inventories, as the myth of motherhood being the ultimate fulfillment for women perpetuates the idea that negative emotions, such as fear, hopelessness, and anxiety, are not commonly experienced. By shedding light on these emotions, we normalize the idea that negative emotions are a normal part of motherhood, including feelings of anxiety, depression, fatigue, exhaustion, and uncertainty. Therefore, we encourage include theoretical and methodological study of this emotional ambivalence as part of the experience of motherhood.

Factor Five, named *Motherhood as a Decision*, incorporated items representing an alternative perspective to the social duty and sense of life viewpoints. These items conveyed beliefs such as a woman's ability to be successful without becoming a mother, that motherhood is not the sole path to starting a family, that a solid couple relationship can exist without children, and that a woman's feminine identity is not solely defined by her reproductive capacity. In essence, these items emphasize that motherhood is ultimately a personal decision.

Overall, the factors newly introduced displayed a comparatively lower level of internal consistency than the original, more traditional factors. This discrepancy can likely be attributed to the varied contexts of motherhood that these new factors explore, as previously mentioned. Another contributing factor could be the limited number of items for each new factor, such as only two items in the *Negative Emotions* factor. However, it is crucial to acknowledge that the original factors, which are deeply rooted in tradition, exhibited higher consistency. This higher consistency may stem from their focus on beliefs that are more established and broadly accepted in our society, particularly those beliefs that correspond with traditional and hegemonic perspectives on motherhood.

The significance of including non-hegemonic beliefs about motherhood lies in several aspects: 1) dispelling the myth of the perfect mother, which imposes demands that jeopardize women's physical, mental, and emotional well-being [22]; 2) reducing the gender gap in academic and professional environments by normalizing the continuation of studies and/or work after becoming a mother [39]; 3) addressing the psychological distress resulting from the pressure to conform to societal expectations of hegemonic motherhood, from the decision-making process to the act of parenting [40]; and 4) alleviating feelings of guilt for deviating from societal norms, whether it be delaying or choosing not to pursue motherhood, prioritizing academic

and/or career success over starting a family [24]. Lastly, it is important to acknowledge that experiencing negative emotions such as fatigue, exhaustion, anxiety, and anguish while navigating motherhood is valid, necessary, and entirely normal [41].

## Study 3

For the final study, we examined the relationship between the new versions of ATGRS and MBS, as well as their association with scores on depression, anxiety, and PPF scales, which serve as measures of mental health.

### Participants

Similarly, as in Study 1 and 2, participants were recruited through the laboratory's social media platforms (Instagram, Facebook, Twitter) and the personal social media accounts of laboratory members from October to December 2022. We invited the general population to participate with the only restriction that they were over 18 years of age. Participants were native-Spanish speakers of Mexican nationality, and all signed a written privacy notice and an informed consent form. Demographic data were anonymously collected including age, last grade of studies, employment status, locality, and marital status. All participants were provided with written information about the study's objectives and procedures. They then signed a privacy notice and informed consent form in accordance with the project's approval from the Ethics Committee of the Faculty of Psychology at UNAM.

### Materials and procedure

**Instruments.** We utilized the extended version of ATGRS from Study 1 and the extended version of MBS from Study 2. Due to the lengthy names of the factors comprising ATGRS, the following abbreviations were employed in Study 3: Factor 1, *Traditional attitudes towards gender roles* (abbreviated as *Traditional factor*); Factor 2, *Favorable attitude towards egalitarian success* (abbreviated as *Egalitarian success factor*); Factor 3, *Favorable attitude towards gender equity* (abbreviated as *Gender equity factor*); and, Factor 4, *Non-traditional attitudes towards gender roles* (abbreviated as *Non-traditional factor)*.

*Beck Depression Inventory [42].* This inventory assesses the frequency and intensity of cognitive-affective and somatic-motivational symptoms of depression over a two-week period. It comprises 21 items that are rated on a Likert-type scale ranging from 0 (no presence of the symptom) to 3 (severe degree of the symptom).

*Beck Anxiety Inventory [43].* This inventory evaluates the frequency with which an individual experiences common somatic and cognitive symptom of anxiety. It consists of 21 items, and respondents rate the items on a Likert-type scale ranging from 0 (not affected) to 3 (severely affected).

*Positive Psychological Functioning Scale (PPF) [44].* This scale evaluates various psychological resources essential for people's psychological well-being including Autonomy, Resilience, Self-esteem, Purpose in life, Enjoyment, Optimism, Curiosity, Creativity, Sense of humor, Mastery of the environment and Vitality. The scale comprises eleven subscales, each consisting of three items. The score can be calculated for each subscale and for the overall total of all items. Participants provided responses on a Likert-type scale from 1 (totally disagree) to 5 (totally agree).

**Data analysis.** Initially, we calculated descriptive statistics for the sociodemographic variables, including measures of central tendency (such as mean and standard deviation or median and interquartile range) and the cumulative percentage frequency of last grade of studies, employment status, locality and marital status. As was expected, people with children were

found to be older and had higher education levels compared to those without children. Consequently, age and education level were incorporated as covariates in association analyses to control for these demographic differences.

To achieve the primary aim of the Study 3, which is to examine the association between the new versions of ATGRS and MBS, as well as their correlation with measures of mental health, we conducted partial correlation analyses using Spearman's non-parametric method to examine the relationships between the different scales while controlling for the effects of covariates, namely age and education level. To account for multiple correlations and reduce the risk of Type I errors, we applied a Bonferroni correction. Additionally, as a complement analysis, using the Ordinary Least Squares (OLS) method, we conducted multiple linear regressions to predict a dependent variable from a set of factors on a scale. For instance, to predict scores on the *Social Duty* factor of the MBS scale, all factors from the ATGRS scale were used as predictor variables. In all cases, the interaction with sex and parental status was calculated, and adjustments were made for age and educational level.

In addition to the main objective, we aimed to assess these variables concerning sex and parental status: ATGRS and MBS in Study 3.2 and measures of mental health in Study 3.4. To achieve this, we utilized the Mann-Whitney test for independent samples to compare the scores of all scales based on parenting status and sex. To mitigate the possibility of Type I errors resulting from multiple comparisons, we applied the Holm correction. All statistical analyses were performed using R Studio software, version 4.1.1 [36], with a significance level set at $p < 0.05$.

Below are the results presented according to specific objectives, including the objective itself, sample details, expected hypotheses, and the corresponding results.

## Study 3.1

As the first objective of this study, we examined the relationship between ATGRS and MBS in the sample of 685 volunteers, with and without children.

According to previous studies which report that people who hold traditional beliefs in one area of society tend to also hold traditional attitudes in other areas, and conversely, non-traditional attitudes are generally prevalent across various social domains [45], we hypothesized that higher scores on traditional attitudes towards gender roles (*Traditional factor*) would be associated with higher scores on the more traditional factors of the MBS scale: *Social duty*, *Sense of life*, and *Hegemonic stereotypes*. Additionally, we expected that higher scores on non-traditional attitudes towards gender roles (*Egalitarian success factor*, *Gender equity factor*, and *Non-traditional factor*) would be related to higher scores in the non-traditional factors of the MBS scale: *Negative emotions* and *Motherhood as a decision*.

**Results.** The sample for Study 3 consisted of 685 participants, including 465 women (67.88%, mean age = 24.95, sd = 9.44, range = 18–67 years), 207 men (30.21%, mean age = 24.43, sd = 9.27, range = 18–71 years), and 13 non-binary individuals (1.89%, mean age = 22.00, sd = 7.06, range = 18–45 years). Table 3 shows demographic data grouped by parenting status and sex.

According to expectations, we observed significant positive correlations between the *Traditional* and *Egalitarian success* factors of ATGRS with the traditional factors of the MBS scale (*Social duty*, *Sense of life*, and *Hegemonic stereotypes*). The *Gender equity* factor of ATGRS showed a significant positive correlation with the non-traditional factors of the MBS scale (*Negative emotions* and *Motherhood as a decision*). Additionally, the *Non-traditional* factor of ATGRS exhibited significant positive correlations with the *Hegemonic stereotypes* and *Negative emotions* factors of MBS. Conversely, we found significant negative correlations between the

**Table 3. Demographic data of participants of study 3 grouped by parenting status and by sex.**

| | PARENTING STATUS | | | | | |
| --- | --- | --- | --- | --- | --- | --- |
| | Without children | | | With children | | |
| | n = 577 | | | n = 108 | | |
| | SEX | | | SEX | | |
| | Women | Men | Non-binary | Women | Men | Non-binary |
| | n = 373 | n = 192 | n = 12 | n = 92 | n = 15 | n = 1 |
| | Mean (sd) | Mean (sd) | Mean (sd) | Mean (sd) | Mean (sd) | Mean (sd) |
| Age | 21.23 (4.04) | 22.64 (6.30) | 20.08 (1.51) | 40.04 (10.01) | 47.27 (11.00) | 45 (—) |
| **Last grade of studies** | **n (%)** | **n (%)** | **n (%)** | **n (%)** | **n (%)** | **n (%)** |
| *none* | 0 | 0 | 0 | 1 (1.09) | 0 | 0 |
| *primary school* | 0 | 0 | 0 | 3 (3.26) | 0 | 0 |
| *secondary school* | 10 (2.62) | 5 (2.60) | 0 | 14 (15.22) | 3 (20.00) | 0 |
| *high school* | 263 (70.51) | 117 (60.94) | 9 (75) | 26 (28.26) | 3 (20.00) | 0 |
| *bachelor's degree* | 92 (24.66) | 66 (34.38) | 3 (25) | 44 (47.83) | 6 (40.00) | 0 |
| *postgraduate degree* | 8 (2.14) | 4 (2.08) | 0 | 4 (4.35) | 3 (20.00) | 1 (100) |
| **Employment status** | **n (%)** | **n (%)** | **n (%)** | **n (%)** | **n (%)** | **n (%)** |
| *Student* | 318 (85.25) | 131 (68.23) | 10 (83.33) | 2 (2.17) | 0 | 0 |
| *Part-time job* | 17 (4.56) | 16 (8.33) | 2 (16.67) | 17 (18.48) | 2 (13.33) | 0 |
| *Full-time job* | 23 (6.17) | 35 (18.23) | 0 | 37 (40.22) | 8 (53.33) | 1 (100) |
| *Unpaid job* | 12 (3.22) | 3 (1.56) | 0 | 29 (31.52) | 1 (6.67) | 0 |
| *Other* | 3 (0.80) | 7 (3.65) | 0 | 7 (7.61) | 4 (26.67) | 0 |
| **Locality** | **n (%)** | **n (%)** | **n (%)** | **n (%)** | **n (%)** | **n (%)** |
| *Mexico City* | 279 (74.80) | 146 (76.04) | 8 (66.67) | 65 (70.65) | 10 (66.67) | 0 |
| *Mexico state* | 77 (20.64) | 41 (21.35) | 3 (25.00) | 21 (22.83) | 3 (20) | 1 (100) |
| *Other state* | 17 (4.56) | 5 (2.60) | 1 (8.33) | 6 (6.52) | 2 (13.33) | 0 |
| **Marital Status** | **n (%)** | **n (%)** | **n (%)** | **n (%)** | **n (%)** | **n (%)** |
| *Single* | 359 (96.25) | 174 (90.63) | 12 (100) | 38 (41.30) | 5 (33.33) | 1 (100) |
| *Married* | 12 (3.22) | 15 (7.81) | 0 | 54 (58.70) | 10 (66.67) | 0 |
| *Engagement* | 2 (0.54) | 3 (1.56) | 0 | 0 | 0 | 0 |
| | **Mean (sd)** | **Mean (sd)** | **Mean (sd)** | **Mean (sd)** | **Mean (sd)** | **Mean (sd)** |
| *Depression* | 19.35 (12.48) | 13.95 (11.29) | 32.83 (15.42) | 12.91 (11.56) | 8.93 (8.94) | 4 |
| *Anxiety* | 19.13 (13.26) | 12.43 (12.27) | 26.92 (14.85) | 12.72 (12.36) | 9.47 (9.83) | 1 |
| *PPF* | 133.14 (29.48) | 142.79 (32.21) | 114.58 (26.09) | 141.14 (36.44) | 155.93 (25.54) | 130 |
| **ATGRS** | **Mean (sd)** | **Mean (sd)** | **Mean (sd)** | **Mean (sd)** | **Mean (sd)** | **Mean (sd)** |
| *Traditional* | 15.96 (4.61) | 19.78 (5.08) | 17.42 (3.40) | 16.83 (5.36) | 19.87 (4.55) | 12 |
| *Egalitarian success* | 57.42 (4.71) | 53.83 (9.05) | 57.33 (3.23) | 55.34 (5.99) | 50.53 (11.04) | 49 |
| *Gender equity* | 8.00 (3.01) | 8.58 (3.42) | 5.92 (2.07) | 9.22 (3.54) | 11.07 (3.37) | 6 |
| *Non-traditional* | 19.51 (5.48) | 19.23 (4.70) | 20.00 (2.92) | 17.52 (5.34) | 17.6 (3.78) | 26 |
| **MBS** | **Mean (sd)** | **Mean (sd)** | **Mean (sd)** | **Mean (sd)** | **Mean (sd)** | **Mean (sd)** |
| *Social Duty* | 0.26 (1.24) | 0.34 (1.49) | 0.08 (0.29) | 0.63 (1.94) | 1.13 (3.02) | 0 |
| *Sense of life* | 2.17 (3.91) | 4.32 (5.68) | 1.50 (1.57) | 7.71 (8.65) | 5.87 (7.8) | 0 |
| *Hegemonic stereotypes* | 9.23 (4.67) | 9.43 (5.34) | 11.17 (5.83) | 9.52 (5.69) | 9.13 (5.84) | 13 |
| *Negative emotions* | 7.83 (2.33) | 7.23 (2.49) | 6.92 (2.27) | 6.61 (3.28) | 5.6 (3.27) | 10 |
| *Motherhood as a decision* | 36.62 (5.60) | 33.96 (8.19) | 37.67 (2.06) | 33.63 (6.84) | 30.53 (8.67) | 40 |

*Gender equity* factor of ATGRS and the traditional factors of the MBS scale (*Social duty*, *Sense of life*, and *Hegemonic stereotypes*), as well as between the *Traditional* and *Egalitarian success* of ATGRS and the *Motherhood as a decision* of MBS (Table 4). These findings were similar when

**Table 4. Spearman partial correlation coefficients between ATGRS and MBS, adjusted for age and educational level.** Positive correlations are indicated in red, while negative correlations are indicated in blue.

| MBS | ATGRS | | | |
|---|---|---|---|---|
| | Traditional | Egalitarian success | Gender equity | Non-traditional |
| | *total sample, n = 685* | | | |
| Social duty | 0.17 | 0.19 | -0.21 | |
| Sense of life | 0.30 | 0.22 | -0.17 | |
| Hegemonic stereotypes | 0.18 | 0.16 | -0.13 | 0.13 |
| Negative emotions | | | 0.18 | 0.14 |
| Motherhood as a decision | -0.19 | -0.25 | 0.28 | |
| | *without children sample, n = 577* | | | |
| Social duty | 0.14 | 0.15 | -0.19 | |
| Sense of life | 0.32 | 0.20 | -0.17 | |
| Hegemonic stereotypes | 0.17 | 0.15 | -0.13 | |
| Negative emotions | | | 0.17 | 0.13 |
| Motherhood as a decision | -0.17 | -0.24 | 0.26 | |
| | *with children sample, n = 108* | | | |
| Social duty | 0.30 | 0.38 | | |
| Sense of life | | 0.31 | | |
| Motherhood as a decision | -0.29 | | 0.37 | |
| | *women without children sample, n = 373* | | | |
| Social duty | 0.17 | | -0.17 | |
| Sense of life | 0.26 | | | |
| Hegemonic stereotypes | 0.20 | | | |
| Negative emotions | | | | 0.17 |
| Motherhood as a decision | | -0.18 | 0.24 | |
| | *men without children sample, n = 192* | | | |
| Social duty | | | -0.25 | |
| Sense of life | 0.28 | 0.36 | -0.23 | |
| Motherhood as a decision | | -0.31 | 0.24 | |
| | *women with children sample, n = 92* | | | |
| Social duty | | 0.40 | | |
| Sense of life | | 0.36 | | |
| Motherhood as a decision | | | 0.39 | |

All coefficients shown in the table were significant at p < 0.05 adjusted using the Bonferroni method.

dividing the total sample into subsamples, that is, in the subsample of individuals without children, including both women and men; however, for the subsample of individuals with children, only some of these correlations remained (Table 4).

Specifically, in the sample of women without children, positive correlations were found between the *Traditional* factor of ATGRS and traditional factors of MBS, and the *Gender equity* factor of ATGRS showed a positive correlation with *Motherhood as a decision*. In men without children, positive correlations were observed between the *Traditional* factor of ATGRS and one traditional factor of MBS (*Sense of life*), as well as between the *Egalitarian success* factor of ATGRS and *Sense of life*. Negative correlations were found between the *Egalitarian success* factor of ATGRS and *Motherhood as a decision*, and between the *Gender equity* factor of ATGRS and traditional factors of MBS (Table 4). For samples with children, significant positive correlations were found only in the sample of women with children, particularly

**Table 5. Multiple lineal model from ATGRS factors for each factor of MBS, adjusted for age and educational level.**

|  |  | Estimate | SE | t-value | p-value |
|---|---|---|---|---|---|
| Social duty | Traditional | 0.05 | 0.01 | 3.67 | p < 0.001 |
|  | Egalitarian success | 0.05 | 0.02 | 2.71 | p < 0.01 |
|  | Gender equity | -0.05 | 0.01 | -5.95 | p < 0.001 |
|  | Non-traditional | -0.05 | 0.02 | -2.11 | p < 0.05 |
| Sense of life | Traditional | 0.24 | 0.05 | 4.53 | p < 0.001 |
|  | Egalitarian success | 0.41 | 0.07 | 6.12 | p < 0.001 |
|  | Gender equity | -0.04 | 0.03 | -1.33 | N.S |
|  | Non-traditional | -0.15 | 0.08 | -1.72 | N.S |
| Hegemonic stereotypes | Traditional | 0.15 | 0.05 | 2.88 | p < 0.01 |
|  | Egalitarian success | 0.19 | 0.06 | 2.98 | p < 0.01 |
|  | Gender equity | -0.04 | 0.03 | -1.29 | N.S |
|  | Non-traditional | 0.01 | 0.08 | 0.07 | N.S |
| Negative emotions | Traditional | -0.02 | 0.02 | -0.85 | N.S |
|  | Egalitarian success | -0.06 | 0.03 | -1.77 | N.S |
|  | Gender equity | 0.06 | 0.02 | 4.02 | p < 0.001 |
|  | Non-traditional | 0.09 | 0.04 | 1.98 | p < 0.05 |
| Motherhood as a decision | Traditional | -0.06 | 0.06 | -1.05 | N.S |
|  | Egalitarian success | -0.39 | 0.08 | -4.84 | p < 0.001 |
|  | Gender equity | 0.28 | 0.04 | 7.25 | p < 0.001 |
|  | Non-traditional | 0.13 | 0.11 | 1.26 | N.S |

N.S., not significant

between the *Egalitarian success* factor of ATGRS and traditional factors of MBS, as well as between the *Gender equity* factor of ATGRS and *Motherhood as a decision* (Table 4).

To confirm and complement these findings, we employed a multiple linear regression model to explore how all factors from the ATGRS scale influence each component of the MBS (Table 5). Moreover, we investigated the interaction of these factors with sex and parenthood status, while controlling for age and educational level. Regarding *Social duty*, the model demonstrated an adjusted $R^2$ value of 0.11 ($F(13, 671) = 7.71$, $p < 0.001$). For the *Sense of life*, the model exhibited an adjusted $R^2$ value of 0.21 ($F(13, 671) = 15.48$, $p < 0.001$). Concerning *Hegemonic stereotypes*, the model indicated an adjusted $R^2$ value of 0.06 ($F(13, 671) = 4.46$, $p < 0.001$). For *Negative emotions*, the model displayed an adjusted $R^2$ value of 0.07 ($F(13, 671) = 5.39$, $p < 0.001$). Regarding *Motherhood as a decision*, the model revealed an adjusted $R^2$ value of 0.17 ($F(13, 671) = 12.29$, $p < 0.001$). No interactions with sex and parenthood status were detected in any model.

The results of the multiple linear regression are also in accordance with the hypotheses. *Social duty*, *Sense of life*, and *Hegemonic stereotypes* as motherhood beliefs were predicted positively by traditional factors of ATGRS (Table 5). Non-traditional factors of ATGRS predicted negatively *Social duty*, while a contrasting pattern was detected for the *Negative emotions* factor of the MBS scale, i.e., it was predicted positively. *Motherhood as a decision* was predicted positively by *Gender equity* and negatively by *Egalitarian success*.

## Study 3.2

Then, we examined whether there were differences in ATGRS and MBS based on sex and parental status. As shown in Table 3 of demographic data, the number of people with and

without children who participated in Study 3 was unequal, n = 108 and n = 577, respectively. To ensure equal sample sizes for the comparison based on Parenting status, we randomly selected 107 individuals without children while maintaining the proportion of women and men from the sample with children (92 women and 15 men) out of a total of 577 individuals. Additionally, for the between-group comparisons analysis based on Parenting status and Sex, we excluded the participant who identified as non-binary in the sex variable, resulting in a total of 107 participants with children.

It has been described that having children has a "traditionalizing" effect on social beliefs [13, 46], we hypothesized that individuals with children would exhibit higher scores in traditional attitudes towards gender roles and in beliefs of motherhood, while those without children would display higher scores in non-traditional attitudes and beliefs. Based on the privileges granted to men over women by hegemonic traditional beliefs about gender roles, we expected men to have higher scores in traditional attitudes towards gender roles and motherhood beliefs, whereas women were anticipated to have higher scores in non-traditional factors in both scales.

**Results.** In demographic data, we detected differences in age and education level between these subgroups; people with children were older than the group without children (W = 10727, p-value < 0.001), and people without children had more years of study, in proportion, to those with children (X-squared = 40.58, df = 5, p-value < 0.001, Table 6).

Regarding the ATGRS, we found significant differences between the parenting status groups in all factors except for the *Traditional* factor. The *Gender equity* factor and *Non-traditional* factor had higher scores in the group without children compared to the group with

**Table 6. Between-group comparisons based on Parenting status on the ATGRS and the MBS (n's equal and standardized by sex).** The p- value was adjusted using the Holm method.

| | PARENTING STATUS | | |
| --- | --- | --- | --- |
| | **Without children** | **With children** | |
| | **n = 107** | **n = 107** | |
| | **Mean (SD)** | **Mean (SD)** | **P-value** |
| **Age** | 21.94 (5.49) | 41.05(10.40) | < 0.001 |
| **Education** | n (%) | n (%) | < 0.001 |
| *none* | 0 | 1 (0.93) | |
| *primary school* | 0 | 3 (2.80) | |
| *secondary school* | 3 (2.80) | 17 (15.88) | |
| *high school* | 73 (68.22) | 29 (27.10) | |
| *bachelor's degree* | 28 (26.16) | 50 (46.72) | |
| *postgraduate degree* | 3 (2.80) | 7 (6.54) | |
| **ATGRS** | **Median (IQR)** | **Median (IQR)** | **P-value** |
| *Traditional* | 16 (5.5) | 17 (7) | N.S. |
| *Egalitarian success* | 8 (4) | 10 (6) | 0.015 |
| *Gender equity* | 60 (3) | 58 (10) | 0.002 |
| *Non-traditional* | 21 (8) | 17 (8) | 0.007 |
| **MBS** | **Median (IQR)** | **Median (IQR)** | **P-value** |
| *Social Duty* | 0 (0) | 0 (0) | N.S. |
| *Sense of life* | 0 (3) | 5 (12) | < 0.001 |
| *Hegemonic stereotypes* | 9 (7) | 9 (9) | N.S. |
| *Negative emotions* | 8 (4) | 7 (5.5) | N.S. |
| *Motherhood as a decision* | 38 (5) | 35 (9.5) | 0.005 |

SD, standard deviation; IQR, interquartile range; N.S., not significant

**Table 7. Between-group comparisons based on Sex on the ATGRS and the MBS in a sample of people without children.** The p- value was adjusted using the Holm method.

| | Sex | | |
|---|---|---|---|
| | **Women** | **Men** | |
| | **n = 192** | **n = 192** | |
| | **Mean (SD)** | **Mean (SD)** | **P-value** |
| **Age** | 21.43 (4.39) | 22.64 (6.30) | N.S. |
| **Education** | n (%) | n (%) | 0.01 |
| *none* | 0 | 0 | |
| *primary school* | 0 | 0 | |
| *secondary school* | 2 (1.04) | 5 (2.60) | |
| *high school* | 144 (75) | 117 (60.93) | |
| *bachelor's degree* | 41 (21.35) | 66 (34.37) | |
| *postgraduate degree* | 5 (2.60) | 4 (2.08) | |
| **ATGRS** | **Median (IQR)** | **Median (IQR)** | **P-value** |
| *Traditional* | 16 (7) | 21 (7.25) | < 0.001 |
| *Egalitarian success* | 8 (4) | 8 (6) | N.S. |
| *Gender equity* | 59 (4) | 58 (11) | 0.001 |
| *Non-traditional* | 21 (7) | 20 (5) | N.S. |
| **MBS** | **Median (IQR)** | **Median (IQR)** | **P-value** |
| *Social Duty* | 0 (0) | 0 (0) | N.S. |
| *Sense of life* | 0 (3.00) | 2 (6.25) | 0.001 |
| *Hegemonic stereotypes* | 9 (7) | 9 (7) | N.S. |
| *Negative emotions* | 8 (3) | 8 (4) | 0.008 |
| *Motherhood as a decision* | 38 (4) | 37 (8) | 0.009 |

SD, standard deviation; IQR, interquartile range; N.S., not significant

children. On the other hand, the *Egalitarian success* factor showed higher scores in the group with children compared to the group without children (middle section of Table 6).

In terms of the MBS, our findings indicated that *Sense of life* factor scored higher in the group with children than in the without children group. Conversely, *Motherhood as a decision* factor had higher scores in the group without children compared to the group with children (lower section of Table 6).

Regarding the between-groups by sex analysis, we only compared people without children because the number of men with and without children was unequal, n = 15 and n = 192 respectively. To equalize the n's of women and men without children, we randomly select a sample of 192 women from the total sample of women without children n = 373. Within these sub-samples, we identified statistically significant differences in years of education (X-squared = 10.03, df = 3, p-value = 0.01) between women and men (Table 7).

Men scored higher in the *Traditional* factor, while women scored higher in the *Gender equity* factor compared to men (middle section of Table 7) on the ATGRS. In the MBS, we observed significant differences in three out of the five factors. Men had higher scores in the *Sense of life* factor, while women had higher scores in *Negative emotions* and *Motherhood as a decision* (lower section of Table 7).

## Study 3.3

Next, in line with our primary objective of examining the association between attitudes toward gender roles, motherhood beliefs, and mental health, we conducted an analysis to explore the

relationship between ATGRS and MBS with mental health outcomes (depression, anxiety, and PPF) in a sample of 685 volunteers. We hypothesize that higher scores in the traditional factors of both scales (ATGRS and MBS) will be positively correlated with higher scores in depression and anxiety, but negatively correlated with PPF. We expected these correlations to remain significant in the subsamples of people with children and in the subsample of women. This is based on previous studies [23, 24, 26] that have reported elevated levels of stress, anger, shame, and guilt, primarily in women with children, due to unrealistic expectations related to traditional concepts of motherhood and gender roles. On the other hand, higher scores in non-traditional attitudes towards gender roles and motherhood beliefs would be correlated with low levels of depression and anxiety, but positively correlated with PPF. We expected these correlations to remain significant in the subsamples of people without children and in the subsample of men.

**Results.** Contrary to expectation, a significant negative correlation between the *Traditional factor* of ATGRS and anxiety was detected. According to hypothesis, we found a significant positive correlation between the *Gender equity factor* of ATGRS and PPF. These findings were consistent across the total sample and the subsample including both women and men without children (Table 8).

Regarding the MBS, we discovered a significant positive correlation between the *Negative emotions* and depression in the total sample, the sample of people without children, and the sample of people with children (Table 8).

To corroborate our findings, we employed a multiple linear regression model to analyze the impact of factors from both the ATGRS and the MBS on mental health measures. This analysis also considered the interaction of these factors with sex and parental status, while adjusting for age and educational level. The model that used ATGRS factors to predict depression yielded an adjusted $R^2$ value of 0.11 ($F(13, 671) = 5.63$, $p < 0.001$). In the case of anxiety, the model

**Table 8. Age and educational level-adjusted Spearman partial correlation coefficients between ATGRS and MBS with mental health (depression, anxiety and PPF).**

| | | Mental health | | |
|---|---|---|---|---|
| | | **Depression** | **Anxiety** | **PPF** |
| **ATGRS** | | *total sample, n = 685* | | |
| | Traditional | | -0.12 | |
| | Gender equity | | | 0.19 |
| | | *without children sample, n = 577* | | |
| | Traditional | | -0.14 | |
| | Gender equity | | | 0.18 |
| | | *women without children sample, n = 373* | | |
| | Egalitarian success | | | -0.17 |
| | Gender equity | | | 0.18 |
| | | *men without children sample, n = 192* | | |
| | Gender equity | | | 0.29 |
| **MBS** | | *total sample, n = 685* | | |
| | Negative emotions | 0.17 | | |
| | | *without children sample, n = 577* | | |
| | Negative emotions | 0.14 | | |
| | | *with children sample, n = 108* | | |
| | Negative emotions | 0.30 | | |

All coefficients shown in the table were significant at p < 0.05 adjusted using the Bonferroni method.

**Table 9. Multiple lineal model from ATGRS factors for depression, anxiety and PPF, adjusted for age and educational level.**

|  |  | Estimate | SE | t-value | p-value |
|---|---|---|---|---|---|
| **Depression** | Traditional | -0.28 | 0.13 | -2.15 | p < 0.05 |
|  | Egalitarian success | 0.03 | 0.16 | 0.17 | N.S. |
|  | Gender equity | -0.01 | 0.07 | -0.18 | N.S. |
|  | Non-traditional | 0.42 | 0.21 | 1.97 | p < 0.05 |
| **Anxiety** | Traditional | -0.38 | 0.14 | -2.74 | p < 0.01 |
|  | Egalitarian success | 0.05 | 0.17 | 0.31 | N.S. |
|  | Gender equity | -0.10 | 0.08 | -1.12 | N.S. |
|  | Non-traditional | 0.05 | 0.02 | 1.01 | N.S. |
| **PPF** | Traditional | 0.36 | 0.32 | 1.09 | N.S. |
|  | Egalitarian success | -0.39 | 0.40 | -0.96 | N.S. |
|  | Gender equity | 0.93 | 0.18 | 4.94 | p < 0.001 |
|  | Non-traditional | -0.89 | 0.54 | -1.64 | N.S. |

N.S., not significant

demonstrated an adjusted $R^2$ value of 0.07 (F(13, 671) = 5.11, p < 0.001). Notably, there was a significant sex-related effect, with women showing higher anxiety scores (estimates = 10.12, SE = 4.69, t = 1.15, p < 0.05). Regarding PPF, the model revealed an adjusted $R^2$ value of 0.08 (F(13, 671) = 5.09, p < 0.001). Traditional factors negatively predicted both depression and anxiety. Conversely, Non-traditional factor were positively associated with depression and *Gender equity* was negatively associated with PPF (Table 9).

In the multiple linear regression model where MBS factors served as predictors for depression, an adjusted $R^2$ value of 0.11 was observed (F(14, 670) = 7.06, p < 0.001). Regarding anxiety, the model showed an adjusted $R^2$ value of 0.07 (F(14, 670) = 4.73, p < 0.001). For PPF, the adjusted $R^2$ value was 0.04 (F(14, 670) = 3.51, p < 0.001). Notably, in these models, no interactions with sex and parental status were evident. Expanding upon the results of the correlation analysis, *Negative emotions*, as a belief related to motherhood, positively predicts both depression and anxiety. However, it had a negative predictive relationship with PPF (Table 10).

## Study 3.4

Finally, to analyze potential differences in mental health variables based on parental status and sex, we applied the same randomization method as in Study 3.2 to ensure equal sample sizes between groups. Although there is no consensus in previous studies regarding mental health outcomes associated with parenthood [47, 48], and considering that parenting has been associated with stress [49], postpartum depression [50], and other mental health symptoms, our hypothesis was that individuals with children would have higher scores in depression, anxiety, and PPF compared to those without children, and that there would be sex differences, with women having higher scores in depression and anxiety but lower scores in PPF compared to men.

**Results.** Contrary to expected, we found that individuals without children had higher scores in depression and anxiety compared to those with children (Table 11). However, higher scores in PPF were observed in individuals with children compared to those without children (Table 11).

According to expectation, women had higher scores in all symptoms of depression and anxiety compared to men. While men had higher scores in PPF scale compared to women (Table 12).

**Table 10. Multiple lineal model from MBS factors for depression, anxiety and PPF, adjusted for age and educational level.**

| | | Estimate | SE | t-value | p-value |
|---|---|---|---|---|---|
| **Depression** | *Social Duty* | 0.58 | 0.36 | 1.58 | N.S. |
| | *Sense of life* | -0.17 | 0.11 | -1.64 | N.S. |
| | *Hegemonic stereotypes* | 0.03 | 0.11 | 0.33 | N.S. |
| | *Negative emotions* | 0.85 | 0.19 | 4.43 | p < 0.001 |
| | *Motherhood as a decision* | -0.06 | 0.11 | -0.63 | N.S. |
| **Anxiety** | *Social Duty* | 0.41 | 0.40 | 1.04 | N.S. |
| | *Sense of life* | -0.11 | 0.11 | -1.03 | N.S. |
| | *Hegemonic stereotypes* | 0.01 | 0.11 | 0.16 | N.S. |
| | *Negative emotions* | 0.41 | 0.21 | 1.96 | p < 0.05 |
| | *Motherhood as a decision* | -0.13 | 0.11 | -1.21 | N.S. |
| **PPF** | *Social Duty* | -0.58 | 0.96 | -0.61 | N.S. |
| | *Sense of life* | 0.01 | 0.27 | 0.06 | N.S. |
| | *Hegemonic stereotypes* | 0.42 | 0.26 | 1.61 | N.S. |
| | *Negative emotions* | -1.41 | 0.50 | -2.79 | p < 0.01 |
| | *Motherhood as a decision* | 0.70 | 0.27 | 2.52 | p < 0.05 |

N.S., not significant

## General discussion

This research aimed to analyze the relationship between attitudes towards gender roles and motherhood beliefs and mental health measures such as depression symptoms, anxiety symptoms, and psychological well-being. To achieve this, we initially expanded two key scales: the scale for Attitudes Towards Gender Roles (ATGRS) in Study 1, and the scale for Motherhood Beliefs (MBS) in Study 2. The modifications included the addition of new items to both scales, updating and aligning the concepts of gender ideology and motherhood beliefs with contemporary perspectives. This approach was designed to mitigate the traditionalist bias inherent in the original versions of these scales. Subsequent to these enhancements, a thorough psychometric validation was conducted for both the ATGRS and MBS scales. The validation process affirmed the overall reliability of the scales. However, it was observed that some of the newly introduced factors exhibited lower levels of consistency compared to the original factors. This difference in consistency may be attributed to the fact that the original factors, which focus on assessing traditional attitudes and beliefs, are more deeply entrenched in our Mexican society [28]. As a result, these traditional factors tend to demonstrate higher consistency, reflecting their more established and widely accepted nature within the societal framework.

This extension of both scales offers two key advantages. The first one is to demonstrate that the study of everyday social phenomena, such as attitudes towards gender roles and

**Table 11. Between-group comparisons based on parenting status on the depression, anxiety and PPF scales (n's equal and standardized by sex).** The p- value was adjusted using the Holm method.

| | PARENTING STATUS | | |
|---|---|---|---|
| | **Without children** | **With children** | |
| | **n = 107** | **n = 107** | |
| | **Median (IQR)** | **Median (IQR)** | **P—value** |
| **Depression** | 17 (14.5) | 10 (17.5) | < 0.001 |
| **Anxiety** | 15 (15) | 9 (16) | < 0.001 |
| **PPF** | 139 (44.5) | 148 (47) | 0.040 |

**Table 12. Between-group comparisons based on sex on the depression, anxiety and PPF scales in a sample of people without children.** The p- value was adjusted using the Holm method.

| | SEX | | |
| --- | --- | --- | --- |
| | Women | Men | |
| | n = 192 | n = 192 | |
| | Median (IQR) | Median (IQR) | P—value |
| *Depression* | 17 (15.5) | 12 (15) | < 0.001 |
| *Anxiety* | 18 (17.25) | 9 (16) | < 0.001 |
| *PPF* | 132.5 (40.50) | 149.5 (43.25) | < 0.001 |

motherhood beliefs, is often approached from a traditional and normative perspective, highlighting the need for studies from non-hegemonic approaches on these social phenomena. The second benefit is to have psychometric instruments that assess both traditional and non-hegemonic positions regarding attitudes towards gender roles and motherhood beliefs, allowing us to better understand their relationship with people's mental health.

Utilizing the expanded versions of both scales, our analysis focused on exploring the potential relationship between attitudes towards gender roles and motherhood beliefs. Consistent with our expectations, we found positive correlations between traditional attitudes towards gender roles and the conventional factors of the MBS, namely *Social Duty*, *Sense of Life*, and *Hegemonic Stereotypes*. This relation was further substantiated through linear regression analysis, which revealed that traditional attitudes towards gender roles positively predicted these traditional MBS factors, even after controlling for age and education level. Our findings are in line with previous studies [45, 51], which indicate that individuals who endorse traditional beliefs in one area are likely to uphold similar attitudes in other social realms. This tendency can potentially restrict opportunities for personal growth, as it often involves conforming to societal expectations, many of which may be unrealistic or unattainable.

On the other hand, the findings pertaining to the non-hegemonic factors showed that individuals with higher scores in non-hegemonic attitudes towards gender roles also exhibited higher scores in non-hegemonic motherhood beliefs. This is evident in the positive association observed between *Gender equity* and *Non-traditional* factor of the ATGRS with the non-traditional factors of the MBS scale: *Negative emotions*, and *Motherhood as a decision*. Moreover, individuals who embraced gender equity and non-traditional gender roles were more inclined to challenge traditional notions of motherhood, especially the concept of motherhood as a *Social Duty*, and were more supportive of viewing motherhood as a personal decision. In contrast, those with traditional gender role attitudes frequently minimized the concept of motherhood as a choice.

However, it is important to recognize that a non-traditional perspective on motherhood can sometimes portray it as a negative or oppressive experience, potentially conflicting with the notion of women's liberation. Ironically, this perspective might also limit the perception of motherhood as a personal choice. Notably, in our study, the *Non-traditional* factor of the ATGRS was not associated with the idea of motherhood as a decision. This observation opens up new avenues for future research to investigate the connection between non-traditional attitudes towards gender roles and the perception of motherhood as a personal choice more deeply.

Next, we examined whether attitudes towards gender roles and motherhood beliefs varied between individuals with and without children. In line with our hypothesis, the results showed that the without children subsample were more inclined towards non-hegemonic beliefs in both gender roles and motherhood. On the other hand, those with children scored higher on

traditional attitudes about gender roles, supporting the idea that having a child primarily had a 'traditionalizing' effect, particularly in women [13, 46]. Likewise, our results reflected that people with children consider that motherhood provides a sense of life, a perspective possibly influenced by their experiences as parents [13, 24, 50]. It is noteworthy that the childless individuals in our study were generally younger than those with children, indicating a potential generational shift in attitudes. This age difference might contribute to a decreased adherence to traditional societal norms. Future research should aim to disentangle the effects of age and parenthood status on gender role and motherhood beliefs more comprehensively, providing further insights into these complex dynamics.

Interestingly, we observed a notable inclination towards traditional beliefs in both the ATGRS and MBS scales among men, as compared to women. Conversely, women scored higher on non-hegemonic beliefs in both scales, on *Gender equity* of the ATGRS and on *Negative emotions* and *Motherhood as a decision* in the MBS. These findings underscore that women oppose traditional gender roles likely because they are the ones who experience the disadvantages and inequalities as a result of gender stereotypes. In contrast, men, benefiting from the privileges offered by patriarchy, do exhibit agreement with traditional gender roles.

These gender-based inequalities extend beyond mere power dynamics and political representation. They manifest in wage disparities, employment opportunities, sexual and reproductive rights, household responsibilities, access to education, and healthcare resources [52]. Women today face the dual challenge of meeting traditional gender role expectations while striving for academic and professional success, in addition to managing domestic duties. This creates a unique set of hurdles for women that their male counterparts, particularly heterosexual men, do not typically encounter [53].

Even when parental roles conform to the gender equity model, there are situations that contribute to the maintenance of traditional parenthood dynamics based on gender. For instance, while women increasingly participate in the workforce, they continue to shoulder the bulk of reproductive work, leading to an often socially invisible double workload [2]. The disparity in time allocation to unpaid work between men and women is still prominent, even in urban areas where women's professional and labor participation has grown. Women spend more than double the time on domestic tasks (such as food preparation, cleaning, caregiving, and voluntary work) compared to men [54]. Furthermore, as men increasingly engage in household chores and parenting [55], they may face discrimination for seeking work flexibility for childcare, along with ridicule and stigmatization from peers [14].

The finding in our study that men favor traditional gender roles raises significant social concerns. This preference contributes to sustaining inequality and hinders the progress of equal rights and opportunities for women. Our results highlight the critical need to challenge and transform gender role hegemony by integrating egalitarian practices into daily life, aiming to eradicate gender-based inequality. However, this transformation is not without its challenges, as evidenced by the correlations we found between the ATGRS and MBS scales and mental health. Addressing these issues is crucial for moving towards a more equitable and just society.

Our initial expectation was that higher scores in traditional gender roles would be linked to increased symptoms of depression and anxiety, as individuals might feel restricted in their expression of behaviors. Contrary to our expectations, we found that individuals with stronger beliefs in traditional gender roles exhibited lower levels of anxiety. This unexpected relationship was further substantiated through linear model analysis, which not only confirmed a negative correlation between traditional gender role beliefs and anxiety but also with depression. Furthermore, the analysis revealed a positive association between non-traditional gender beliefs and depression. These results suggest a complex interplay between gender role beliefs

and mental health, challenging our initial assumptions and pointing towards nuanced dynamics in how these beliefs impact psychological well-being.

The *Congruence Hypothesis* possibly explains our findings; it suggests that individuals adhering to stereotypical gender roles (e.g., highly masculine men and highly feminine women) generally experience better mental health. This is attributed to their behaviors and attitudes being in line with cultural norms, which may offer protection from social pressures [6] and related mental health challenges. This hypothesis is particularly applicable in societies where individuals prioritize external social demands over their own interests and desires [29]. In such societies, including Mexico, adherence to established norms can offer a sense of security and stability, thereby alleviating the stress and challenges tied to societal expectations.

Regarding psychological well-being, as measured by the PPF scale, our study found that higher endorsements of gender equity attitudes were associated with improved psychological well-being across all subgroups, with the exception of people with children. This exception may be attributed to the heightened social pressure faced by parents when attempting to adopt gender equity attitudes, owing to the strong presence of traditional beliefs about childcare. This pressure can create a conflict for parents between embracing progressive attitudes and adhering to entrenched societal norms regarding parenting.

Women who step away from traditional motherhood roles to pursue employment often encounter mental health challenges due to the dual demands of professional and domestic responsibilities. Parenthood can potentially diminish the mental health benefits usually derived from other life experiences, such as career development, particularly for women with young children [56]. The pressure to meet the ideal of a perfect mother can be overwhelming, and even without fully internalizing this standard, women may still face adverse mental health effects, including stress, anxiety, and reduced self-efficacy.

In contrast, men who deviate from conventional paternalistic roles—often characterized by emotional inexpressiveness and the expectation to be the primary providers [29]—and participate more actively in childcare and nurturing may experience increased satisfaction and fulfillment. This shift from traditional roles can be particularly rewarding [12]. This dynamic aligns with observations in countries with high gender equality, where mothers are more susceptible to depression, while in countries with lower gender equality, unemployed women often face the opposite trend. For men, the influence of children on mental health appears to be less significant, irrespective of the country's level of gender equality [57]. This highlights the differential impact of parenting roles on mental health across genders and cultural contexts.

Our research also identified a significant positive correlation between believing that motherhood can generate negative emotions such as fear and anxiety, with a higher level of depression. This finding persisted in the subsamples with and without children. Additionally, linear model analysis revealed that this factor, negative emotions, positively predicted higher rates of depression and anxiety, while inversely predicting psychological well-being.

In addition, we observed that a strong belief in motherhood as a personal choice correlated with higher levels of positive psychological functioning. This indicates that attitudes of gender equity and viewing motherhood as a personal decision positively impact psychological health. These findings underscore the significance of advocating for predominantly egalitarian practices in daily life, contributing to promote parenthood practices not grounded on traditional gender mandates and societal impositions. Such endeavors may play a pivotal role in enhancing the mental health of individuals.

Our findings suggest that it is not simply the status of being a parent that affects mental health, but rather the underlying beliefs about parenting and motherhood. The mental health outcomes of individuals seem to be significantly influenced by their attitudes and beliefs

regarding motherhood, highlighting the complex interplay between personal beliefs and psychological well-being.

Finally, we evaluated mental health measures in relation to sex and parental status. Contrary to our initial hypotheses, we found that individuals with children reported lower levels of depression and anxiety, and higher levels of psychological well-being, compared to those without children. This observation is consistent with the parental well-being model [47], which posits that despite the challenges of parenting, fulfilling the societal role of a parent can have a positive impact on mental health. It's also noted that parents often experience a heightened sense of purpose, given their responsibility for someone else, which can foster resilience and a more optimistic outlook [13].

In essence, our findings suggest that individuals with children are likely to perceive a stronger sense of purpose in their lives, i.e. sense of life. For instance, people with children view motherhood or parenthood not just as a life-enriching experience but also as a social duty, a perspective not as prevalent among those without children. Additionally, in a cultural context like Mexico's, where parenthood is highly esteemed and embodies collectivist values, it might serve as a mental health protective factor. This cultural esteem for parenthood could alleviate social pressures and distribute the demands of child-rearing more evenly within extended family and community networks.

Our findings regarding mental health scores in women and men without children align with previous research. Women had higher depression and anxiety scores compared to men, while men had higher psychological well-being scores compared to women. There is extensive evidence supporting the reasons behind the lower mental health scores in women, including societal pressure related to bodily appearance, behavior, and motherhood stereotypes, as well as issues like *machismo*, gender violence, and the burden of double work shifts involving both employment outside the home and domestic responsibilities such as childcare, housework, and caring for sick or elderly family members [53, 58, 59].

Another possible explanation for lower depression and anxiety scores in men could be their adherence to traditional masculinity stereotypes, where characteristics such as strength, bravery, and maintaining control are valued. Mahalik and Dagirmanjian [60] propose that stigma and the threat to their masculine identity can negatively influence men's willingness to seek help for depression and sadness. Seidler et al. [61] conducted a review of recommendations for involving men in psychological treatment and found that most of the articles they reviewed took a universal perspective on masculinity, which often resulted in poor outcomes for men by encouraging them to avoid seeking psychological help. The authors emphasize the impact of gender norms on clinical engagement and the results of psychological treatment for men.

In conclusion, our study revealed a significant association between higher levels of traditional attitudes towards gender roles and traditional motherhood beliefs, while conversely, higher levels of non-traditional attitudes towards gender roles were correlated with non-traditional motherhood beliefs. Interestingly, we observed that traditional attitudes toward gender roles were associated with lower anxiety and depression scores, while non-traditional attitudes were associated with higher levels of depression. Furthermore, individuals who embraced non-traditional attitudes towards both gender roles and motherhood beliefs tended to exhibit better psychological well-being. These trends were consistent across various demographics, including men and women, and those with and without children. Notably, our findings revealed that women generally showed lesser alignment with traditional attitudes towards both gender roles and motherhood beliefs compared to men. However, women reported higher rates of depression and anxiety, alongside lower psychological well-being scores, than their male counterparts. This highlights the significant influence that traditional cultural norms about gender roles and motherhood have on women's mental health. These norms contribute

to increased anxiety and depression among women and adversely affect their overall psychological well-being, underscoring the need for a deeper understanding and reevaluation of these traditional constructs in society.

## Limitations

One significant limitation of our study was the restricted range of instruments available for precisely measuring motherhood beliefs. Moreover, despite the large sample sizes, there was an overrepresentation of women (67.88%), individuals of younger age (averaging 21 years old), and those without children. This uneven representation introduces a potential bias in the data, underscoring the need for caution in interpreting the results. For future studies, it would be beneficial to aim for a sample more reflective of the Mexican population's demographics, where women constitute 51.2% and the median age is 29 years, as per INEGI's 2021 data [62]. Furthermore, the proportion of participants with children in our study was relatively small (15.77%). Although we assessed the interaction of gender and parental status in Study 3, the findings should be interpreted with care due to the aforementioned biases in sample representativeness regarding these variables. Further research is warranted to delve into the influence of social constructs on gender roles and motherhood, utilizing larger and more diverse samples of men and women with children. Incorporating additional variables, such as marital status, family structure, and pre-parenthood attitudes towards gender roles, could also enhance the depth of our understanding of the attitudes and beliefs examined in this study.

## Acknowledgments

We would like to express our appreciation to all people that participated in our study.

## Author Contributions

**Conceptualization:** Anabel Claudia Aceves-Gómez, Azalea Reyes-Aguilar.

**Data curation:** Anabel Claudia Aceves-Gómez.

**Formal analysis:** Maribel Delgado-Herrera, Azalea Reyes-Aguilar.

**Funding acquisition:** Azalea Reyes-Aguilar.

**Investigation:** Maribel Delgado-Herrera, Anabel Claudia Aceves-Gómez.

**Methodology:** Maribel Delgado-Herrera, Anabel Claudia Aceves-Gómez.

**Project administration:** Azalea Reyes-Aguilar.

**Resources:** Azalea Reyes-Aguilar.

**Supervision:** Azalea Reyes-Aguilar.

**Visualization:** Maribel Delgado-Herrera.

**Writing – original draft:** Maribel Delgado-Herrera.

**Writing – review & editing:** Maribel Delgado-Herrera, Anabel Claudia Aceves-Gómez, Azalea Reyes-Aguilar.

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
