## [Decision Letter · Decision Letter 0]

30 Oct 2023

PONE-D-23-22025Relationship between gender roles, motherhood beliefs and mental healthPLOS ONE

Dear Dr. Reyes-Aguilar,

Thank you for submitting your manuscript to PLOS ONE. After careful consideration, we feel that it has merit but does not fully meet PLOS ONE’s publication criteria as it currently stands. Therefore, we invite you to submit a revised version of the manuscript that addresses the points raised during the review process.

We look forward to receiving your revised manuscript.

Kind regards,

Sergi Fàbregues

Academic Editor

PLOS ONE

Journal Requirements:

**Additional Editor Comments:**

This paper has been reviewed by two experts in the field who recognized the contribution of the paper to the literature but, at the same time, identified a number of major issues that need to be addressed by the author before submitting the paper for a second round of peer review. Among the issues raised by the reviewers, with which I agree, are the need to add key literature on the topic of the study (reviewer 1 suggests some references), to provide a more elaborate contextualization of the topic of the study and key concepts, to better explain some elements of the methodology, to clarify various parts of the results, and to acknowledge some study limitations. Although reviewer 2 questions the current organization of the article, I think it already works well in its current form. Please address the reviewers' comments and write a response to each of them, indicating how you have addressed them (or provide a justification if you have not).

Reviewers' comments:

Reviewer's Responses to Questions

**Comments to the Author**

1. Is the manuscript technically sound, and do the data support the conclusions?

Reviewer #1: Partly

Reviewer #2: Partly

2. Has the statistical analysis been performed appropriately and rigorously? 

Reviewer #1: No

Reviewer #2: Yes

3. Have the authors made all data underlying the findings in their manuscript fully available?

Reviewer #1: No

Reviewer #2: Yes

4. Is the manuscript presented in an intelligible fashion and written in standard English?

Reviewer #1: Yes

Reviewer #2: Yes

5. Review Comments to the Author

Reviewer #1: In this ambitious series of studies, this paper expanded two measures, the Attitudes Towards Gender Roles Scale (ATGRS) and the Motherhood Beliefs Scale (MBS), to capture more contemporary and diverse perspectives on gender roles and motherhood beliefs. After assessing the psychometric properties of both extended versions of these scales (in Studies 1 and 2), their associations with three indicators of mental health (i.e., depressive symptoms, anxiety symptoms, and positive psychological functioning) were assessed (Study 3). In addition to examining the relationship between the newly developed versions of ATGRS and MBS and mental health, sex and parental status differences in these associations were also investigated. Overall the study produced updated versions of ATGRS and MBS that capture both traditional and non-hegemonic attitudes towards gender roles and motherhood. These scales have the potential for use in future research on these topics. Regarding results from Study 3, which focused on mental health, the authors found evidence that espousing more traditional views on gender roles and motherhood were associated with better mental health (i.e., fewer depressive and anxiety symptoms, higher scores on positive psychological functioning). The study has several strengths, including the careful consideration of new items for two previously validated scales and establishing that these measures are related to mental health. However, there are some limitations and areas of needed clarification for the study in its current form. I have attempted to organize my comments by section (e.g., Introduction, Discussion) and study (1, 2, or 3).

1) Introduction: The discussion of gender roles, motherhood beliefs, and mental health is missing some key literature on the topic. Perhaps these papers can help the authors to better contextualize their study findings in the Introduction and Discussion sections:

Evenson, R. J., & Simon, R. W. (2005). Clarifying the relationship between parenthood and depression. Journal of Health and Social Behavior, 46(4), 341–358. https://doi.org/10.1177/002214650504600403

Leupp, K. (2017). Depression, work and family roles, and the gendered life course. Journal of Health and Social Behavior, 58(4), 422–441. https://doi.org/10.1177/0022146517737309

Vespa, J. (2009). Gender ideology construction: A life course and intersectional approach. Gender & Society, 23(3), 363–387. doi:10.1177/0891243209337507.

2) Recruitment of study participants for Studies 1, 2, and 3: It appears that participants were recruited from social media platforms (e.g., Facebook, Instagram). Given the recruitment strategy, the sample may be biased towards younger and more tech-savvy individuals. Could the authors speak to this issue?

3) Study 1: Though the mean and SD for age was reported, what was the age range?

4) Study 2: I appreciate the authors’ expansion of items for the Motherhood Belief Scale. However, the low alpha reliability score (.54) for Hegemonic Stereotypes is a concern, particularly because the alpha reliability scores for the other factors are .77 or higher. Why do the authors believe the alpha was particularly low for hegemonic stereotypes, and what might future research do to further refine this factor of the MBS?

5) Study 2: Related to the previous point, the alpha reliability scores for the newly added factors (i.e., hegemonic stereotypes, negative emotions, and motherhood as a decision) are all lower compared to the previously established factors (Social duty, sense of life). Why might this be the case?

6) Study 2: The “Negative emotions” factor for the new version of MBS only has two items. Why weren’t the range of emotions captured in the items developed for this factor? As currently constructed, the negative emotions factor only captures “scary” and “anxiety”, but in the discussion of this factor, the authors note that “fatigue” and “anguish” are also emotions associated with motherhood. It seems like a missed opportunity to not include other items that capture other “negative” emotions that have been linked to motherhood.

7) Study 3: The seemingly strong positive correlation between age and parental status are concerning because it is difficult to determine if the results are driven by parental status or actual age. For instance, in Table 3, respondents without children are in their early 20s in terms of age while those with children are in their 40s (on average). The authors note that both parental status and age/generation have been shown to be related to gender role and motherhood beliefs. Given the significant overlap between age and parental status in these data, it seems difficult to determine if being a parent or being younger/older is driving differences in gender roles and motherhood beliefs.

8) Study 3: It’s not clear why Spearmon’s non-parametric method was used to assess the association between gender role/motherhood beliefs and mental health. Why would the authors not use OLS regression analysis that would allow for adjustments for age, education level, and other sociodemographic characteristics? With OLS regression, the authors could also test for gender and parental status moderation by testing for statistical interactions between gender and gender role beliefs/motherhood beliefs, and statistical interactions between parental status and gender role beliefs/motherhood beliefs.

9) Study 3: The randomization technique for making the sample sizes conquerable for individuals with and without children is concerning (e.g., this is mentioned on page 35, lines 643-645 page 36, lines 663-665, page 40, lines 732-734). This seems to be a “waste” of perfectly analyzable data. At the least, the authors could report how the results compare when the full sample is utilized? Also, is there a citation or reference for establishing precedence in the quantitative methodological literature for this decision to remove a significant portion (N =470) of the sample from the analysis?

10) Though three measures of mental health were included in Study 3, other mental health problems associated with externalizing behaviors (e.g., substance use, anti-social behavioral disorders) are more common among men, as masculinity is associated with increased risk-taking behavior. Perhaps the associations between gender role beliefs and mental health would look different for men if other measures of mental health were assessed (e.g., substance abuse disorders). This is not necessarily a limitation of the study, but should be noted as an area for future research. For citations on this topic, see:

Smith, D. T., Mouzon, D. M., & Elliott, M. (2018). Reviewing the assumptions about men’s mental health: An exploration of the gender binary. American journal of men's health, 12(1), 78-89.

Rosenfield, Sarah, and Dawne Mouzon. (2013). "Gender and mental health." Handbook of the sociology of mental health pp. 277-296.

11) Discussion: The discussion section is very well-written and thoughtful. However, the issue of employment continued to come into focus. Are there any data on employment status and/or occupation for Study 3? The authors are theorizing that gender roles/motherhood beliefs will be dependent on gender, parental status, and employment status. The literature they have cited also seems to strongly suggest this is the case as well.

12) Discussion: Given that the data were collected in Mexico and in the Spanish language, could the authors speak more about how this particular context (i.e., Mexico) may be relevant in terms of discussing the study findings? They start to do this somewhat on page 46 (when discussing Hopcroft and McLaughlin), but more could be stated about gender dynamics within Mexico and how those social dynamics could help with interpretation of the study results.

13) Discussion: Do the authors envision that the newly developed scales can be utilized in non-Spanish speaking countries and would there be any necessary adjustments to the scale if asked outside of Mexico?

Overall, the study makes an important contribution to the literature, particularly regarding advancements in the measurement of gender role attitudes as well as motherhood beliefs. Additional clarity on issues raised above will help to further clarify and contextualize the study findings.

Reviewer #2: The article presents a relevant topic with interesting findings. However, some issues prevent me from accepting it in its current form.

The article is divided into a general introduction, studies 1, 2, and 3, and a general discussion. Studies 1 and 2 are related to the validation of two scales used in the methodology of study 3. This structure makes it challenging to understand the research. Given that the abstract emphasizes study 3, I recommend concentrating on this study alone within the abstract and integrating a summarized version of the scale validation process in the methodology section. This reorganization will enhance clarity and overall understanding of the research.

The central concern of this study is related to the sample. In the abstract, a sample of 3,927 people is cited. However, a substantial portion of this sample was involved in validating the measurement instruments, leaving 685 participants for the study concerning the relationship between gender roles, conceptions of motherhood, and mental health. After stratifying by gender, this number dropped to 214 participants. This leads to questions about the study's validity and representativeness, which must be addressed in the text.

It is essential to provide details regarding the characteristics of the 685 participants. The majority of them were childless women with high levels of education. How representative are they of the broader population? How might selection biases be at play, and how were they controlled for? Furthermore, participants without children had a median age of 20, while those with children had a median age of 40. How might this age difference influence the study's results? Could generational factors confound the findings? Notably, the participants with children included only 15 men (constituting 14% of the group). How was this low number addressed?

Concerning the study's conceptual framework, there's a need to develop the concept of gender more comprehensively within the introduction, as explicitly addressed in the second paragraph. Mexico has a robust history of gender theory, including studies on masculinities. It is advisable to include references to the following:

* The concept of gender is multifaceted, historically contingent, and culturally shaped, and, therefore, is not static.

* One of the prevailing definitions of gender is the hegemonic model standard in Western societies, characterized by the sexual division of labor, which this study refers to as traditional gender roles. While the article references hegemonic gender roles, it lacks an explicit definition of sexual division of labor.

Moreover, it's essential to clarify that, in this article, the term "masculinity" pertains to the hegemonic model of masculinity. There are various models of masculinity, and this specificity should be underscored.

The introduction's fourth and fifth paragraphs discuss the challenges men and women face as they navigate the roles assigned by the sexual division of labor. It would be beneficial to provide context, highlighting that women have entered the workforce without a concurrent redistribution of responsibilities in the domestic sphere, which has implications for women's health.

From the introduction's 8th to 13th paragraph, methodological details are discussed, which would be more appropriate for the methods section rather than the introduction.

Lastly, in the final paragraph of the introduction, there is a notable absence of a reference to the gaps this study aims to address concerning the relationships between gender roles, conceptions of motherhood, and mental health within the study population.

As for the methods section, in addition to the previously mentioned general comments, the following is requested:

* To include more extensive information about the participants, such as their geographical diversity within Mexico.

* To include a synthesis of studies 1 and 2. The synthesis should demonstrate that the scales are appropriate for measuring what they intend to measure and valid for the Mexican context. If the scales were validated in populations with specific characteristics (e.g., more women, childless, educated, etc.), whether they are valid for the broader Mexican population should be clarified.

* Explain the study's independent, covariate, and outcome variables, and detail the instruments used to measure them. While the instruments used are listed, it is crucial to establish what constructs these scales are measuring, their origins, and whether they have been validated for use in Mexico.

* Link each analysis to its specific objective.

Within the results section:

* Include a descriptive table presenting the overall results of the measurement instruments used in the sample.

* Regarding the first objective (examining the relationship between the Attitudes Towards Gender Roles Scale and the Motherhood Beliefs Scale, stratified by parental status), the results of the correlations between the scales are described. However, a comparison of the dimensions of these correlations needs to be included. Are specific correlations more significant than others?

* For the second objective (examining differences in ATGRS and MBS based on sex and parental status), it is noted that the sample was adjusted without controlling for age and education levels in people with and without children, as they were not the primary focus of the study. This explanation is considered insufficient. It is requested to include a clarification regarding this issue, especially regarding the low representation of men in the sample.

* The research's central focus is the third objective, which examines the relationship between ATGRS and MBS with mental health. However, the absence of a gender-based comparison and its implications for the results still need to be addressed.

In the general discussion, it is critical to recognize that the research methodology employed does not enable the establishment of causal relationships, for instance, about the idea that having a child primarily had a 'traditionalizing' effect. Even if the results show that childless people are more likely to hold non-hegemonic beliefs while individuals with children obtained higher scores in traditional attitudes about gender roles, it is unclear if there is a reverse causality. It could be that people with children have always been more conservative and, consequently, have children. Alternatively, generational variables may confound the results, given that the childless group is around 20 years old, while the group with children is around 40. Finally, in the last sentence of the discussion, it is possible to state that the study shows a correlation between these variables. Still, it cannot establish a causal relationship between them.

6. PLOS authors have the option to publish the peer review history of their article (what does this mean?). If published, this will include your full peer review and any attached files.

Reviewer #1: No

Reviewer #2: No

---

## [Author Response · Author response to Decision Letter 0]

28 Dec 2023

Dear Mr. Sergi Fàbregues,

I am writing to inform you that we had carefully reviewed the feedback provided by the academic editor and reviewers regarding our manuscript titled "Relationship between gender roles, motherhood beliefs, and mental health," which was submitted to PLOS ONE under Manuscript No. PONE-D-23-22025.

We had taken the reviewers' comments carefully and had made the necessary revisions to address the points raised during the review process. The revisions were carried out with the aim of improving the overall quality and rigor of our research.

In response to your request, we have included the following items in the revised manuscript:

A rebuttal letter that responds to each point raised by the academic editor and reviewers. This letter is included as a separate file labeled 'Response_to_Reviewers'.

A marked-up copy of the manuscript that highlights changes made to the original version. This file is labeled 'Revised_Manuscript_with_Track_Changes'

An unmarked version of the revised paper without tracked changes. This file is labeled 'Manuscript.'

We appreciate your time and effort in reviewing our work and thank you for providing us with this opportunity to make necessary improvements. We are committed to delivering a high-quality manuscript that aligns with the standards of PLOS ONE.

For reviewers: 

Reviewer #1: In this ambitious series of studies, this paper expanded two measures, the Attitudes Towards Gender Roles Scale (ATGRS) and the Motherhood Beliefs Scale (MBS), to capture more contemporary and diverse perspectives on gender roles and motherhood beliefs. After assessing the psychometric properties of both extended versions of these scales (in Studies 1 and 2), their associations with three indicators of mental health (i.e., depressive symptoms, anxiety symptoms, and positive psychological functioning) were assessed (Study 3). In addition to examining the relationship between the newly developed versions of ATGRS and MBS and mental health, sex and parental status differences in these associations were also investigated. Overall the study produced updated versions of ATGRS and MBS that capture both traditional and non-hegemonic attitudes towards gender roles and motherhood. These scales have the potential for use in future research on these topics. Regarding results from Study 3, which focused on mental health, the authors found evidence that espousing more traditional views on gender roles and motherhood were associated with better mental health (i.e., fewer depressive and anxiety symptoms, higher scores on positive psychological functioning). The study has several strengths, including the careful consideration of new items for two previously validated scales and establishing that these measures are related to mental health. However, there are some limitations and areas of needed clarification for the study in its current form. I have attempted to organize my comments by section (e.g., Introduction, Discussion) and study (1, 2, or 3).

1) Introduction: The discussion of gender roles, motherhood beliefs, and mental health is missing some key literature on the topic. Perhaps these papers can help the authors to better contextualize their study findings in the Introduction and Discussion sections:

Evenson, R. J., & Simon, R. W. (2005). Clarifying the relationship between parenthood and depression. Journal of Health and Social Behavior, 46(4), 341–358. https://doi.org/10.1177/002214650504600403

Leupp, K. (2017). Depression, work and family roles, and the gendered life course. Journal of Health and Social Behavior, 58(4), 422–441. https://doi.org/10.1177/0022146517737309

Vespa, J. (2009). Gender ideology construction: A life course and intersectional approach. Gender & Society, 23(3), 363–387. doi:10.1177/0891243209337507.

Response: We appreciate the suggestion and have incorporated additional literature into the introduction and discussion sections, using the references suggested by Evenson & Simon (2005), Leupp (2017), and Vespa (2009) to better contextualize our findings.

Now this integration can be read in the Introduction, line 76:

Vespa [11] contends that gender ideology is dynamic, evolving over time as individuals encounter diverse social environments that introduce them to gender expectations related to aspects like marriage, parenthood, and work. Various determinants, such as race, age of children, education, family type, and pre-parenthood attitudes toward gender roles, contribute to this organic nature [11–13]. Fatherhood, for instance, often entails the expectation of being the primary breadwinner, whereas motherhood encompasses responsibilities related to childcare, nurturing, and housework [14]. Nevertheless, diverse experiences of parenthood influence gender ideology, shaped by the specific gender expectations individuals encounter. For married couples, parenthood tends to be associated with less egalitarian gender ideologies, irrespective of gender and race [11]. Conversely, some studies suggest that parenthood may reinforce traditional gender role stereotypes when compared to childless individuals [13]. Moreover, there is evidence indicating that individuals may develop more traditional attitudes and behaviors regarding gender roles once they become parents [15–17].

In Discussion, it reads (line 895):

Women who step away from traditional motherhood roles to pursue employment often encounter mental health challenges due to the dual demands of professional and domestic responsibilities. Parenthood can potentially diminish the mental health benefits usually derived from other life experiences, such as career development, particularly for women with young children [56]. The pressure to meet the ideal of a perfect mother can be overwhelming, and even without fully internalizing this standard, women may still face adverse mental health effects, including stress, anxiety, and reduced self-efficacy.

In contrast, men who deviate from conventional paternalistic roles—often characterized by emotional inexpressiveness and the expectation to be the primary providers [29]—and participate more actively in childcare and nurturing may experience increased satisfaction and fulfillment. This shift from traditional roles can be particularly rewarding [12]. This dynamic aligns with observations in countries with high gender equality, where mothers are more susceptible to depression, while in countries with lower gender equality, unemployed women often face the opposite trend. For men, the influence of children on mental health appears to be less significant, irrespective of the country's level of gender equality [57]. This highlights the differential impact of parenting roles on mental health across genders and cultural contexts.

2) Recruitment of study participants for Studies 1, 2, and 3: It appears that participants were recruited from social media platforms (e.g., Facebook, Instagram). Given the recruitment strategy, the sample may be biased towards younger and more tech-savvy individuals. Could the authors speak to this issue?

Response: We have acknowledged the observation about recruitment and we are aware that our sample has a bias towards younger people and people who use technology. While our recruitment method allowed us to efficiently reach a diverse pool of participants, we recognize its limitations. However, we will make this issue explicit in the limitations section (line 980), as follows:

One significant limitation of our study was the restricted range of instruments available for precisely measuring motherhood beliefs. Moreover, despite the large sample sizes, there was an overrepresentation of women (67.88%), individuals of younger age (averaging 21 years old), and those without children. This uneven representation introduces a potential bias in the data, underscoring the need for caution in interpreting the results. For future studies, it would be beneficial to aim for a sample more reflective of the Mexican population's demographics, where women constitute 51.2% and the median age is 29 years, as per INEGI's 2021 data [62]. Furthermore, the proportion of participants with children in our study was relatively small (15.77%). Although we assessed the interaction of gender and parental status in Study 3, the findings should be interpreted with care due to the aforementioned biases in sample representativeness regarding these variables.

3) Study 1: Though the mean and SD for age was reported, what was the age range?

Response: We have added the following information about the age range in Study 1 (line 235): 

The sample for Study 1 consisted of 2,677 participants, including 1,742 females (65.07%), 927 males (34.62%), and eight non-binary individuals (0.29%). The participants had a mean age of 23.36 years with a standard deviation of 7.76, and with an age range of 18 - 50 years.

In Study 3, it reads (line 582):

The sample for Study 3 consisted of 685 participants, including 465 women (67.88%, mean age = 24.95, sd = 9.44, range = 18 - 67 years), 207 men (30.21%, mean age = 24.43, sd = 9.27, range = 18 - 71 years), and 13 non-binary individuals (1.89%, mean age = 22.00, sd = 7.06, range = 18 - 45 years).

4) Study 2: I appreciate the authors’ expansion of items for the Motherhood Belief Scale. However, the low alpha reliability score (.54) for Hegemonic Stereotypes is a concern, particularly because the alpha reliability scores for the other factors are .77 or higher. Why do the authors believe the alpha was particularly low for hegemonic stereotypes, and what might future research do to further refine this factor of the MBS?

Response: Thank you for your thoughtful feedback on our manuscript. We appreciate your positive acknowledgment of the expansion of items for the Motherhood Belief Scale (MBS) in Study 2. Regarding your concern about the low alpha reliability score (.54) for the Hegemonic Stereotypes factor, we believe the lower alpha may be attributed to some aspects such as heterogeneity of context associated to motherhood (i.e. responsibility, happiness, breeding, needs, work performance) that include the six items in this factor. Although hegemonic stereotypes of motherhood are expressed in all these contexts, there may be differences among them. However, both exploratory and confirmatory factor analyses group these items into a single factor that we refer to by its common factor: Hegemonic Stereotypes. In the ongoing development of this scale, we will consider adding more items for each context to enhance the reliability of the scale. 

For now, we began with a general search of the scientific literature on motherhood, and from there, the items were generated, as stated in the manuscript. It was only after the exploratory and confirmatory factor analysis that these items were grouped into a factor we named: Hegemonic Stereotypes.

In study 2: Instruments and procedure, it reads (line 373):

To enhance the scale, we introduced additional items that more thoroughly examine various aspects of motherhood while avoiding gender stereotypes. These new items were derived from an extensive review of theoretical and scientific literature on motherhood from 2000 to 2021. This review led to the identification of a wide range of definitions, concepts, and beliefs about motherhood, which were then adapted into Likert-type statements. As a result, 16 new items were added to the MBS, covering areas such as female identity, decision-making, lifestyle, success, life plans, and emotions including love, frustration, pain, and passion. For example, new items included 'Maternity is a woman's decision' and 'Motherhood can generate anxiety,' as detailed in Table 2. Participants completed this extended version of the MBS via a Google Form.

In study 2: Results, it reads (line 426):

After confirming that the 28 items were allocated into five factors, we named and ordered them based on the degree of traditionalism, from most to least traditional: Factor 1, Social duty, included four items from original-MBS; Factor 2, Sense of life, comprised eight items from original-MBS; Factor 3, Hegemonic stereotypes comprised six new items; Factor 4, Negative emotions, included two new items; and, Factor 5, Motherhood as a decision, encompassed eight new items (Table 2).

In study 2: Discussion of study 2, it reads (line 447): 

The third factor, named Hegemonic Stereotypes, aims to underscore the predominant beliefs about motherhood that reinforce the exclusive responsibility of mothers in child-rearing. It reflects societal expectations of full-time dedication as a criterion for being deemed a good mother, emphasizing complete responsibility and happiness in motherhood, absolute commitment to breeding, and the act of relinquishing one's own needs and the professional field. Although Hegemonic Stereotypes are expressed across these diverse contexts (responsibility, parenting style, professional work, personal needs). The inclusion of various contexts in the items of this factor may have contributed to its lower consistency. Future efforts could focus on refining this factor by either narrowing down the included contexts or incorporating more items for each context.

5) Study 2: Related to the previous point, the alpha reliability scores for the newly added factors (i.e., hegemonic stereotypes, negative emotions, and motherhood as a decision) are all lower compared to the previously established factors (Social duty, sense of life). Why might this be the case?

Response: We appreciate your attention to detail and your thoughtful comments. Regarding the alpha reliability scores for the newly added factors in Study 2, namely Hegemonic stereotypes, Negative emotions, and Motherhood as a decision, we acknowledge that they are lower compared to the previously established factors, Social duty and Sense of life.

As we mentioned in response to a similar point earlier, the lower alpha reliability scores for these newly added factors may stem from the limited number of items comprising in each factor, the diverse contexts explored by the items within each factor, which might have introduced more complexity and, consequently, resulted in lower internal consistency. We are aware of this limitation, and in our future work, we plan to explore ways to refine these factors, whether by narrowing down the included contexts or by adding more items for each factor. 

In another possible consideration, which we do not rule out, it could be that these new factors, representing less hegemonic or traditional beliefs about motherhood, are present with less strength or consistency compared to traditionalist beliefs, that is, the perception of motherhood as a duty and a sense of life for women. We added this possible explanation in the General discussion of the manuscript, you can read it at line 781:

The modifications included the addition of new items to both scales, updating and aligning the concepts of gender ideology and motherhood beliefs with contemporary perspectives. This approach was designed to mitigate the traditionalist bias inherent in the original versions of these scales. Subsequent to these enhancements, a thorough psychometric validation was conducted for both the ATGRS and MBS scales. The validation process affirmed the overall reliability of the scales. However, it was observed that some of the newly introduced factors exhibited lower levels of consistency compared to the original factors. This difference in consistency may be attributed to the fact that the original factors, which focus on assessing traditional attitudes and beliefs, are more deeply entrenched in our Mexican society [28] . As a result, these traditional factors tend to demonstrate higher consistency, reflecting their more established and widely accepted nature within the societal framework.

6) Study 2: The “Negative emotions” factor for the new version of MBS only has two items. Why weren’t the range of emotions captured in the items developed for this factor? As currently constructed, the negative emotions factor only captures “scary” and “anxiety”, but in the discussion of this factor, the authors note that “fatigue” and “anguish” are also emotions associated with motherhood. It seems like a missed opportunity to not include other items that capture other “negative” emotions that have been linked to motherhood.

Response: We appreciate your insightful observation regarding the limited number of items in the "Negative emotions" factor in the new version of the MBS. We completely agree with your concern and acknowledge that the current construction of this factor captures only a subset of negative emotions associated with motherhood, specifically "scary" and "anxiety." We recognize that emotions such as "fatigue" and "anguish" are essential components of the emotional experience related to motherhood and were not explicitly included in our current set of items.

We want to emphasize that this limitation was not overlooked but rather a deliberate decision in the initial phase of scale development. The intention was to start with a more focused exploration and later expand the factor by including additional items that could better capture the diverse range of negative emotions associated with motherhood. We see your point as an opportunity to enhance the scale's comprehensiveness, and we are actively planning to include more items in this factor in our future work.

By expanding the range of emotions covered, we aim to create a more nuanced and comprehensive understanding of the emotional aspects of motherhood, considering both positive and negative dimensions. Your feedback aligns with our vision for refining the MBS, and we sincerely appreciate your valuable input.

In the manuscript, we discuss possible reasons for differences in alpha reliability and highlight the importance of future research to address these issues. You can read it at line 476: 

Overall, the factors newly introduced displayed a comparatively lower level of internal consistency than the original, more traditional factors. This discrepancy can likely be attributed to the varied contexts of motherhood that these new factors explore, as previously mentioned. Another contributing factor could be the limited number of items for each new factor, such as only two items in the Negative Emotions factor. However, it is crucial to acknowledge that the original factors, which are deeply rooted in tradition, exhibited higher consistency. This higher consistency may stem from their focus on beliefs that are more established and broadly accepted in our society, particularly those beliefs that correspond with traditional and hegemonic perspectives on motherhood.

7) Study 3: The seemingly strong positive correlation between age and parental status are concerning because it is difficult to determine if the results are driven by parental status or actual age. For instance, in Table 3, respondents without children are in their early 20s in terms of age while those with children are in their 40s (on average). The authors note that both parental status and age/generation have been shown to be related to gender role and motherhood beliefs. Given the significant overlap between age and parental status in these data, it seems difficult to determine if being a parent or being younger/older is driving differences in gender roles and motherhood beliefs.

Response: Thank you for your thoughtful consideration of our study. We acknowledge the concern you raised about the potential confounding effects of age and parental status on the observed results in Study 3. In response to this concern, we have conducted additional analyses to disentangle the influence of age and parental status on gender roles and motherhood beliefs. Specifically, we performed a lineal regression analysis with age and education as a covariate to control for its potential effects. The results of these analyses are presented in the revised version of the manuscript, and we discuss the nuanced interplay between age and parental status in shaping attitudes toward gender roles and motherhood. Now you can read it at line 836:

It is noteworthy that the childless individuals in our study were generally younger than those with children, indicating a potential generational shift in attitudes. This age difference might contribute to a decreased adherence to traditional societal norms. Future research should aim to disentangle the effects of age and parenthood status on gender role and motherhood beliefs more comprehensively, providing further insights into these complex dynamics. 

8) Study 3: It’s not clear why Spearman's non-parametric method was used to assess the association between gender role/motherhood beliefs and mental health. Why would the authors not use OLS regression analysis that would allow for adjustments for age, education level, and other sociodemographic characteristics? With OLS regression, the authors could also test for gender and parental status moderation by testing for statistical interactions between gender and gender role beliefs/motherhood beliefs, and statistical interactions between parental status and gender role beliefs/motherhood beliefs.

Response: We thank you for your proposed statistical analysis. In the new version of the manuscript, we performed, as a complementary analysis, using the ordinary least squares (OLS) method, multiple linear regressions to predict as dependent variables the scores of all factors on one scale (e.g. Social Duty on the MBS scale) and all factors on the other scale (in this example, the ATGRS factors) as predictor variables. In all models, the interaction with sex and parental status was calculated, and adjustments were made for age and educational level. You can read the results of these analyses from Study 3.1 at line 617:

To confirm and complement these findings, we employed a multiple linear regression model to explore how all factors from the ATGRS scale influence each component of the MBS (Table 5). Moreover, we investigated the interaction of these factors with sex and parenthood status, while controlling for age and educational level. Regarding Social duty, the model demonstrated an adjusted R2 value of 0.11 (F(13, 671) = 7.71, p < 0.001). For the Sense of life, the model exhibited an adjusted R2 value of 0.21 (F(13, 671) = 15.48, p < 0.001). Concerning Hegemonic stereotypes, the model indicated an adjusted R2 value of 0.06 (F(13, 671) = 4.46, p < 0.001). For Negative emotions, the model displayed an adjusted R2 value of 0.07 (F(13, 671) = 5.39, p < 0.001). Regarding Motherhood as a decision, the model revealed an adjusted R2 value of 0.17 (F(13, 671) = 12.29, p < 0.001). No interactions with sex and parenthood status were detected in any model.

Table 5. Multiple lineal model from ATGRS factors for each factor of MBS, adjusted for age and educational level. 

 Estimate SE t-value p-value

Social duty Traditional 0.05 0.01 3.67 p < 0.001

 Egalitarian success 0.05 0.02 2.71 p < 0.01

 Gender equity -0.05 0.01 -5.95 p < 0.001

 Non-traditional -0.05 0.02 -2.11 p < 0.05

Sense of life Traditional 0.24 0.05 4.53 p < 0.001

 Egalitarian success 0.41 0.07 6.12 p < 0.001

 Gender equity -0.04 0.03 -1.33 N.S

 Non-traditional -0.15 0.08 -1.72 N.S

Hegemonic stereotypes Traditional 0.151 0.05 2.88 p < 0.01

 Egalitarian success 0.194 0.06 2.98 p < 0.01

 Gender equity -0.04 0.03 -1.29 N.S

 Non-traditional 0.01 0.08 0.074 N.S

Negative emotions Traditional -0.02 0.02 -0.85 N.S

 Egalitarian success -0.06 0.03 -1.77 N.S

 Gender equity 0.06 0.02 4.02 p < 0.001

 Non-traditional 0.09 0.04 1.98 p < 0.05

Motherhood as a decision Traditional -0.06 0.06 -1.05 N.S

 Egalitarian success -0.39 0.08 -4.84 p < 0.001

 Gender equity 0.28 0.04 7.25 p < 0.001

 Non-traditional 0.13 0.11 1.26 N.S

N.S., not significant

 The results of the multiple linear regression are also in accordance with the hypotheses. Social duty, Sense of life, and Hegemonic stereotypes as motherhood beliefs were predicted positively by traditional factors of ATGRS (Table 5). Non-traditional factors of ATGRS predicted negatively Social duty, while a contrasting pattern was detected for the Negative emotions factor of the MBS scale, i.e., it was predicted positively. Motherhood as a decision was predicted positively by Gender equity and negatively by Egalitarian success.

And from Study 3.3 at line 723:

To corroborate our findings, we employed a multiple linear regression model to analyze the impact of factors from both the ATGRS and the MBS on mental health measures. This analysis also considered the interaction of these factors with sex and parental status, while adjusting for age and educational level. The model that used ATGRS factors to predict depression yielded an adjusted R2 value of 0.11 (F(13, 671) = 5.63, p < 0.001). In the case of anxiety, the model demonstrated an adjusted R2 value of 0.07 (F(13, 671) = 5.11, p < 0.001). Notably, there was a significant sex-related effect, with women showing higher anxiety scores (estimates = 10.12, SE = 4.69, t = 1.15, p < 0.05). Regarding PPF, the model revealed an adjusted R2 value of 0.08 (F(13, 671) = 5.09, p < 0.001). Traditional factors negatively predicted both depression and anxiety. Conversely, Non-traditional factor were positively associated with depression and Gender equity was negatively associated with PPF (Table 9).

 Table 9. Multiple lineal model from ATGRS factors for depression, anxiety and PPF, adjusted for age and educational level. 

 Estimate SE t-value p-value

Depression Traditional -0.28 0.13 -2.15 p < 0.05

 Egalitarian success 0.03 0.16 0.17 N.S.

 Gender equity -0.01 0.07 -0.18 N.S.

 Non-traditional 0.42 0.21 1.97 p < 0.05

Anxiety Traditional -0.38 0.14 -2.74 p < 0.01

 Egalitarian success 0.05 0.17 0.31 N.S.

 Gender equity -0.10 0.08 -1.12 N.S.

 Non-traditional 0.05 0.02 1. N.S.

PPF Traditional 0.36 0.32 1.09 N.S.

 Egalitarian success -0.39 0.40 -0.96 N.S.

 Gender equity 0.93 0.18 4.94 p < 0.001

 Non-traditional -0.89 0.54 -1.64 N.S.

N.S., not significant

In the multiple linear regression model where MBS factors served as predictors for depression, an adjusted R2 value of 0.11 was observed (F(14, 670) = 7.06, p < 0.001). Regarding anxiety, the model showed an adjusted R2 value of 0.07 (F(14, 670) = 4.73, p < 0.001). For PPF, the adjusted R2 value was 0.04 (F(14, 670) = 3.51, p < 0.001). Notably, in these models, no interactions with sex and parental status were evident. Expanding upon the results of the correlation analysis, Negative emotions, as a belief related to motherhood, positively predicts both depression and anxiety. However, it had a negative predictive relationship with PPF (Table 10).

Table 10. Multiple lineal model from MBS factors for depression, anxiety and PPF, adjusted for age and educational level. 

 Estimate SE t-value p-value

Depression Social Duty 0.58 0.36 1.58 N.S.

 Sense of life -0.17 0.11 -1.64 N.S.

 Hegemonic stereotypes 0.03 0.11 0.33 N.S.

 Negative emotions 0.85 0.19 4.43 p < 0.001

 Motherhood as a decision -0.06 0.11 -0.63 N.S.

Anxiety Social Duty 0.41 0.40 1.04 N.S.

 Sense of life -0.11 0.11 -1.03 N.S.

 Hegemonic stereotypes 0.01 0.11 0.16 N.S.

 Negative emotions 0.41 0.21 1.96 p < 0.05

 Motherhood as a decision -0.13 0.11 -1.21 N.S.

PPF Social Duty -0.58 0.96 -0.61 N.S.

 Sense of life 0.01 0.27 0.06 N.S.

 Hegemonic stereotypes 0.42 0.26 1.61 N.S.

 Negative emotions -1.41 0.50 -2.79 p < 0.01

 Motherhood as a decision 0.70 0.27 2.52 p < 0.05

N.S., not significant

9) Study 3: The randomization technique for making the sample sizes conquerable for individuals with and without children is concerning (e.g., this is mentioned on page 35, lines 643-645 page 36, lines 663-665, page 40, lines 732-734). This seems to be a “waste” of perfectly analyzable data. At the least, the authors could report how the results compare when the full sample is utilized? Also, is there a citation or reference for establishing precedence in the quantitative methodological literature for this decision to remove a significant portion (N =470) of the sample from the analysis?

Response: Thank you for your thorough review and insightful comments. We appreciate your concern regarding the undersampling issue raised in Study 3. We have carefully considered your suggestion and made significant revisions to address this concern.

In the revised manuscript, we have included an updated Table 3 that presents the scores of all variables measured in the research, stratified by sex and parenting status. The patterns observed in the data, including the higher scores in traditional beliefs for individuals with children, are now statistically substantiated through analyses conducted on the respective subsamples.

Regarding the decision to create subsamples, we aimed to ensure equal sample sizes across groups to facilitate a fair comparison, particularly in the presence of noticeable disparities in the sizes of some subsamples (e.g., the sample with children, n = 108, compared to the sample without children, n = 577). While we acknowledge the limitation of uneven sample distribution, we believe this approach allowed us to conduct meaningful analyses and draw valuable insights from the available data.

We have added a detailed discussion of this limitation in the revised Limitations section (line 986), acknowledging the trade-off between achieving equitable sample sizes and the potential loss of information. We believe that this transparency will enhance the interpretability of our study's findings. In limitations section, it reads (line 986):

Furthermore, the proportion of participants with children in our study was relatively small (15.77%). Although we assessed the interaction of gender and parental status in Study 3, the findings should be interpreted with care due to the aforementioned biases in sample representativeness regarding these variables. Further research is warranted to delve into the influence of social constructs on gender roles and motherhood, utilizing larger and more diverse samples of men and women with children. Incorporating additional variables, such as marital status, family structure, and pre-parenthood attitudes towards gender roles, could also enhance the depth of our understanding of the attitudes and beliefs examined in this study.

10) Though three measures of mental health were included in Study 3, other mental health problems associated with externalizing behaviors (e.g., substance use, anti-social behavioral disorders) are more common among men, as masculinity is associated with increased risk-taking behavior. Perhaps the associations between gender role beliefs and mental health would look different for men if other measures of mental health were assessed (e.g., substance abuse disorders). This is not necessarily a limitation of the study, but should be noted as an area for future research. For citations on this topic, see:

Smith, D. T., Mouzon, D. M., & Elliott, M. (2018). Reviewing the assumptions about men’s mental health: An exploration of the gender binary. American journal of men's health, 12(1), 78-89.

Rosenfield, Sarah, and Dawne Mouzon. (2013). "Gender and mental health." Handbook of the sociology of mental health pp. 277-296.

Response: We sincerely appreciate your insightful suggestion and your recognition of potential avenues for further research. Exploring additional measures of mental health, particularly those associated with externalizing behaviors such as substance use and anti-social behavioral disorders, is an excellent point. We acknowledge the importance of considering gender-specific aspects in mental health outcomes, as highlighted in the provided references. In our future research, we will carefully address this aspect to enrich our understanding of the subject. Thank you for your valuable input, and we look forward to enhancing the depth of our methodology in this regard.

11) Discussion: The discussion section is very well-written and thoughtful. However, the issue of employment continued to come into focus. Are there any data on employment status and/or occupation for Study 3? The authors are theorizing that gender roles/motherhood beliefs will be dependent on gender, parental status, and employment status. The literature they have cited also seems to strongly suggest this is the case as well.

Response: We express our gratitude for the question regarding employment and occupation data. In the new version of the manuscript, we included more information regarding the demographic data of the sample. You can review that information in new Table 3 (line 586). 

Table 3. Demographic data of participants of study 3 grouped by parenting status and by sex.

 PARENTING STATUS

 Without children With children

 n=577 n=108

 SEX SEX

 Women Men Non-binary Women Men Non-binary

 n=373 n=192 n=12 n=92 n=15 n=1

 Mean (sd) Mean (sd) Mean (sd) Mean (sd) Mean (sd) Mean (sd)

Age 21.23 (4.04) 22.64 (6.30) 20.08 (1.51) 40.04 (10.01) 47.27 (11.00) 45 (--)

Last grade of studies n (%) n (%) n (%) n (%) n (%) n (%)

none 0 0 0 1 (1.09) 0 0

primary school 0 0 0 3 (3.26) 0 0

secondary school 10 (2.62) 5 (2.60) 0 14 (15.22) 3 (20.00) 0

high school 263 (70.51) 117 (60.94) 9 (75) 26 (28.26) 3 (20.00) 0

bachelor’s degree 92 (24.66) 66 (34.38) 3 (25) 44 (47.83) 6 (40.00) 0

postgraduate degree 8 (2.14) 4 (2.08) 0 4 (4.35) 3 (20.00) 1 (100)

Occupation n (%) n (%) n (%) n (%) n (%) n (%)

Student 318 (85.25) 131 (68.23) 10 (83.33) 2 (2.17) 0 0

Part-time job 17 (4.56) 16 (8.33) 2 (16.67) 17 (18.48) 2 (13.33) 0

Full-time job 23 (6.17) 35 (18.23) 0 37 (40.22) 8 (53.33) 1 (100)

Unpaid job 12 (3.22) 3 (1.56) 0 29 (31.52) 1 (6.67) 0

Other 3 (0.80) 7 (3.65) 0 7 (7.61) 4 (26.67) 0

Locality n (%) n (%) n (%) n (%) n (%) n (%)

Mexico City 279 (74.80) 146 (76.04) 8 (66.67) 65 (70.65) 10 (66.67) 0

Mexico state 77 (20.64) 41 (21.35) 3 (25.00) 21 (22.83) 3 (20) 1 (100)

Other state 17 (4.56) 5 (2.60) 1 (8.33) 6 (6.52) 2 (13.33) 0

Marital Status n (%) n (%) n (%) n (%) n (%) n (%)

Single 359 (96.25) 174 (90.63) 12 (100) 38 (41.30) 5 (33.33) 1 (100)

Married 12 (3.22) 15 (7.81) 0 54 (58.70) 10 (66.67) 0

Engagement 2 (0.54) 3 (1.56) 0 0 0 0

 Mean (sd) Mean (sd) Mean (sd) Mean (sd) Mean (sd) Mean (sd)

Depression 19.35 (12.48) 13.95 (11.29) 32.83 (15.42) 12.91 (11.56) 8.93 (8.94) 4

Anxiety 19.13 (13.26) 12.43 (12.27) 26.92 (14.85) 12.72 (12.36) 9.47 (9.83) 1

PPF 133.14 (29.48) 142.79 (32.21) 114.58 (26.09) 141.14 (36.44) 155.93 (25.54) 130

ATGRS Mean (sd) Mean (sd) Mean (sd) Mean (sd) Mean (sd) Mean (sd)

Traditional 15.96 (4.61) 19.78 (5.08) 17.42 (3.40) 16.83 (5.36) 19.87 (4.55) 12

Egalitarian success 57.42 (4.71) 53.83 (9.05) 57.33 (3.23) 55.34 (5.99) 50.53 (11.04) 49

Gender equity 8.00 (3.01) 8.58 (3.42) 5.92 (2.07) 9.22 (3.54) 11.07 (3.37) 6

Non-traditional 19.51 (5.48) 19.23 (4.70) 20.00 (2.92) 17.52 (5.34) 17.6 (3.78) 26

MBS Mean (sd) Mean (sd) Mean (sd) Mean (sd) Mean (sd) Mean (sd)

Social Duty 0.26 (1.24) 0.34 (1.49) 0.08 (0.29) 0.63 (1.94) 1.13 (3.02) 0

Sense of life 2.17 (3.91) 4.32 (5.68) 1.50 (1.57) 7.71 (8.65) 5.87 (7.8) 0

Hegemonic stereotypes 9.23 (4.67) 9.43 (5.34) 11.17 (5.83) 9.52 (5.69) 9.13 (5.84) 13

Negative emotions 7.83 (2.33) 7.23 (2.49) 6.92 (2.27) 6.61 (3.28) 5.6 (3.27) 10

Motherhood as a decision 36.62 (5.60) 33.96 (8.19) 37.67 (2.06) 33.63 (6.84) 30.53 (8.67) 40

In addition, we added to the manuscript a more detailed discussion of the relationship between gender roles, motherhood beliefs and employment as it reads (line 855):

For instance, while women increasingly participate in the workforce, they continue to shoulder the bulk of reproductive work, leading to an often socially invisible double workload [2]. The disparity in time allocation to unpaid work between men and women is still prominent, even in urban areas where women's professional and labor participation has grown. Women spend more than double the time on domestic tasks (such as food preparation, cleaning, caregiving, and voluntary work) compared to men [54]. Furthermore, as men increasingly engage in household chores and parenting [55], they may face discrimination for seeking work flexibility for childcare, along with ridicule and stigmatization from peers [14].

12) Discussion: Given that the data were collected in Mexico and in the Spanish language, could the authors speak more about how this particular context (i.e., Mexico) may be relevant in terms of discussing the study findings? They start to do this somewhat on page 46 (when discussing Hopcroft and McLaughlin), but more could be stated about gender dynamics within Mexico and how those social dynamics could help with interpretation of the study results.

Response: We sincerely appreciate the suggestion to broaden the discussion in the Mexican context. In fact, guided by your comments, we decided that it was important to add information on the Mexican context in the introduction of the paper. In the new version of the introduction, we added (line 109):

In Mexico, family and gender roles adhere strictly to societal expectations, imposing significant pressure to conform. A central cultural aspect is affiliative obedience, where both men and women embody the distinct feature of Mexican culture: self-sacrifice. This implies a belief that prioritizing the needs of others over one's own is more important [28]. Mexicans frequently prioritize external social demands over their personal interests and desires [29].

The traditional Mexican family serves as an illustrative context to understand how gender roles are instilled and acquired in Mexico. According to Diaz-Loving et al. [28], Mexican families are guided by two main axes: a) the power and absolute supremacy of the father, and b) the love and essential sacrifice of the mother. This family structure is mirrored in normative traits of Mexicans, including the emphasis on obedience, the fear of family dishonor, and the magnification of virginity even among adolescents. Moreover, predominant beliefs in this culture include machismo, fear of parental authority, and the importance of respect.

Nevertheless, in Mexico there is a growing diversification of family structures, contemporary meanings related to the family are going beyond the exclusivity of the nuclear biparental model and there is greater openness and tolerance regarding possible alternative forms of coexistence and personal and family development [30]. Within this diversity, coupled with the collectivist character of Mexican society [31], is the participation of networks of women and extended family in the care of children, a term known as othermothers [32]. This could facilitate some conditions such as women having a support network that allows them to occupy other professional and work roles or to have time for self-care, reducing the parental burden.

In addition, in the General discussion section, we incorporated specific perspectives on gender dynamics in Mexico. It can be read at line 883:

This hypothesis is particularly applicable in societies where individuals prioritize external social demands over their own interests and desires [29]. In such societies, including Mexico, adherence to established norms can offer a sense of security and stability, thereby alleviating the stress and challenges tied to societal expectations.

And at line 937:

Additionally, in a cultural context like Mexico's, where parenthood is highly esteemed and embodies collectivist values, it might serve as a mental health protective factor. This cultural esteem for parenthood could alleviate social pressures and distribute the demands of child-rearing more evenly within extended family and community networks.

13) Discussion: Do the authors envision that the newly developed scales can be utilized in non-Spanish speaking countries and would there be any necessary adjustments to the scale if asked outside of Mexico?

Response: Thank you for raising the question regarding the applicability of our newly developed scales in non-Spanish speaking countries. We believe that these scales encompass a broad spectrum of conceptualizations about gender roles and motherhood beliefs that could potentially be relevant in diverse contexts (e.g. Western countries). However, we acknowledge that gender roles and motherhood beliefs, being social constructs, are dynamic and strongly influenced by the sociocultural context. As we noted at lines 164 (for the ATGRS): 

The authors conducted an exploratory study prior to developing the scale to ensure its content was both culturally sensitive and had face validity. They used open questions to collect diverse conceptions -encompassing behaviors, traits, beliefs, etc.- that men and women have regarding their gender identity [33].

At line 374 for the MBS:

These new items were derived from an extensive review of theoretical and scientific literature on motherhood from 2000 to 2021. This review led to the identification of a wide range of definitions, concepts, and beliefs about motherhood, which were then adapted into Likert-type statements.

In response to your query, we have incorporated additional information in the introduction section (line 55) highlighting the key aspects of the context in which these scales were developed. We emphasize the importance of considering the sociocultural nuances specific to Mexico, and we note that adjustments may be necessary if the scales are applied in different cultural settings.

In Western societies, gender ideology is molded by traditional gender roles defined by the sexual division of labor, where specific tasks are strictly assigned based on gender [1]. Women are typically tasked with reproductive work within the home and family, involving domestic management, infrastructure maintenance, and the care of family members. Conversely, men are expected to engage in productive work, contributing to the production of public goods and services, with economic retribution and social recognition [2].

 At line 115, we argue that Mexico is a society where family values are well established and highly esteemed:

According to Diaz-Loving et al. [28], Mexican families are guided by two main axes: a) the power and absolute supremacy of the father, and b) the love and essential sacrifice of the mother. 

This is important and relevant for certain factors: Traditional attitudes towards gender roles and Non-traditional attitudes towards gender roles in the ATGRS and Sense of life and Social duty in the MBS.

Additionally, we emphasize that Mexico is currently undergoing a transformation in social dynamics, including shifts in family roles (line 121)

in Mexico there is a growing diversification of family structures, contemporary meanings related to the family are going beyond the exclusivity of the nuclear biparental model and there is greater openness and tolerance regarding possible alternative forms of coexistence and personal and family development [30]

Within this context lies the need of including the new factors that circumscribe to these transformations: Favorable attitude towards egalitarian success, Favorable attitude towards gender equity for the ATGRS and Motherhood as a decision, Negative Emotions, and Hegemonic Stereotypes factors of the MBS.

Overall, the study makes an important contribution to the literature, particularly regarding advancements in the measurement of gender role attitudes as well as motherhood beliefs. Additional clarity on issues raised above will help to further clarify and contextualize the study findings.

Response: We are deeply thankful for the thoughtful comments and constructive suggestions provided by Reviewer #1. Your insights have significantly contributed to the improvement of our manuscript.

Reviewer #2: The article presents a relevant topic with interesting findings. However, some issues prevent me from accepting it in its current form.

The article is divided into a general introduction, studies 1, 2, and 3, and a general discussion. Studies 1 and 2 are related to the validation of two scales used in the methodology of study 3. This structure makes it challenging to understand the research. Given that the abstract emphasizes study 3, I recommend concentrating on this study alone within the abstract and integrating a summarized version of the scale validation process in the methodology section. This reorganization will enhance clarity and overall understanding of the research.

Response: We sincerely appreciate your recommendation regarding the reorganization of the investigation. We agree that the summary emphasizes Study 3, which could have made it difficult to understand the structure of the research. However, we have reasons to approach the article with this structure, but thanks to your comments we realized that we did not make them clear in the previous version of the manuscript.

Although Study 3 is the central point of the entire investigation, if we had carried out the study with the psychological instruments validated in our country to date, we would have obtained biased results and with limited information, specifically with predominantly traditional ideas and biased towards gender stereotypes.

For this reason, it was strictly necessary to extend and psychometrically validate the instruments before conducting study 3. We consider that both validations deserve an independent section to highlight and discuss 1) the lack of more inclusive measurement instruments on these topics and 2) the importance to know the new factors that the confirmatory factor analysis reveals in each one, exposing the topics that have not yet been explored and the gaps in the research that remains to be carried out.

Following your recommendations, we propose to clarify these reasons in the following way. Firstly, we write a new abstract, it reads (line 26):

Gender roles, as social constructs, play a significant role in shaping individuals' beliefs and attitudes, influencing various aspects of life, including perceptions and expectations surrounding motherhood. These beliefs, acquired through culture and society, can have an impact on our mental well-being. This research consists of three independent studies. In the first and second studies, we extended the Attitudes Towards Gender Roles Scale and Motherhood Beliefs Scale and performed psychometric validation through exploratory and confirmatory factor analysis. The aim of including additional items in both scales was to update these attitudes and beliefs in Mexican culture to avoid the traditionalist bias in both instruments. Finally, the third study examined the relationship between the new versions of both scales and symptoms of depression, anxiety, and Positive Psychological Functioning as indicators of mental health in women and men with and without children. Our findings revealed a significant association between higher levels of traditional attitudes towards gender roles and traditional motherhood beliefs, as well as between non-traditional attitudes towards gender roles and non-traditional beliefs about motherhood. Interestingly, we observed that traditional attitudes toward gender roles were associated with lower anxiety and depression scores, while non-traditional attitudes were associated with higher levels of depression. Furthermore, individuals who embraced non-traditional attitudes towards both gender roles and motherhood beliefs tended to exhibit better psychological well-being in all subsamples. Additionally, women generally showed lesser alignment with traditional attitudes towards both gender roles and motherhood beliefs compared to men. However, women reported higher rates of depression and anxiety, along with lower psychological well-being scores, than their male counterparts. This highlights the significant influence that traditional cultural norms about gender roles and motherhood have on women's mental health, underscoring the need for a deeper understanding and reevaluation of these traditional constructs in society.

Secondly, we will clarify this point in more detail in the last part of Introduction, it reads (line 134):

To test this hypothesis, the first step would be to have updated instruments that probe attitudes towards gender roles today, otherwise the results between gender roles and mental health could be biased.

The present research was divided into three independent studies. In the first and second studies, we extended the Attitudes Towards Gender Roles Scale (ATGRS) [33] and Motherhood Beliefs Scale (MBS) [34] and performed psychometric validation through exploratory and confirmatory factor analysis. The aim of including additional items in both scales was to update attitudes and beliefs in Mexican culture in order to avoid the traditionalist bias in both instruments. 

Third, we will highlight in the General discussion the relevance of the extension and validation of these instruments, it reads (line 779):

To achieve this, we initially expanded two key scales: the scale for Attitudes Towards Gender Roles (ATGRS) in Study 1, and the scale for Motherhood Beliefs (MBS) in Study 2. The modifications included the addition of new items to both scales, updating and aligning the concepts of gender ideology and motherhood beliefs with contemporary perspectives. This approach was designed to mitigate the traditionalist bias inherent in the original versions of these scales. Subsequent to these enhancements, a thorough psychometric validation was conducted for both the ATGRS and MBS scales. The validation process affirmed the overall reliability of the scales. 

The central concern of this study is related to the sample. In the abstract, a sample of 3,927 people is cited. However, a substantial portion of this sample was involved in validating the measurement instruments, leaving 685 participants for the study concerning the relationship between gender roles, conceptions of motherhood, and mental health. After stratifying by gender, this number dropped to 214 participants. This leads to questions about the study's validity and representativeness, which must be addressed in the text.

Response: We sincerely believe that the study will be interpreted better after incorporating your suggestions. In this new version of the abstract, we mention that there are three studies and the importance of carrying them out independently, it reads (line 29):

This research consists of three independent studies. In the first and second studies, we extended the Attitudes Towards Gender Roles Scale and Motherhood Beliefs Scale and performed psychometric validation through exploratory and confirmatory factor analysis. The aim of including additional items in both scales was to update these attitudes and beliefs in Mexican culture to avoid the traditionalist bias in both instruments. Finally, the third study examined the relationship between the new versions of both scales and symptoms of depression, anxiety, and Positive Psychological Functioning as indicators of mental health in women and men with and without children. 

And we aborded this issue on the Limitations section (line 981), as follows:

Moreover, despite the large sample sizes, there was an overrepresentation of women (67.88%), individuals of younger age (averaging 21 years old), and those without children. This uneven representation introduces a potential bias in the data, underscoring the need for caution in interpreting the results. For future studies, it would be beneficial to aim for a sample more reflective of the Mexican population's demographics, where women constitute 51.2% and the median age is 29 years, as per INEGI's 2021 data [62]. Furthermore, the proportion of participants with children in our study was relatively small (15.77%). Although we assessed the interaction of gender and parental status in Study 3, the findings should be interpreted with care due to the aforementioned biases in sample representativeness regarding these variables. Further research is warranted to delve into the influence of social constructs on gender roles and motherhood, utilizing larger and more diverse samples of men and women with children. Incorporating additional variables, such as marital status, family structure, and pre-parenthood attitudes towards gender roles, could also enhance the depth of our understanding of the attitudes and beliefs examined in this study.

It is essential to provide details regarding the characteristics of the 685 participants. The majority of them were childless women with high levels of education. How representative are they of the broader population? How might selection biases be at play, and how were they controlled for? Furthermore, participants without children had a median age of 20, while those with children had a median age of 40. How might this age difference influence the study's results? Could generational factors confound the findings? Notably, the participants with children included only 15 men (constituting 14% of the group). How was this low number addressed?

Response: Regarding your query about the characteristics of the participants, we acknowledge and fully understand the importance of providing a detailed description of the sample (Table 3). The participants in our Study 3, numbering 685, primarily consisted of childless women with elevated levels of education. We agree that this demographic composition raises questions about the representativeness of our sample in relation to the broader population. To address this concern, we have now included additional information on the characteristics of our participants, emphasizing the prevalence of childless women with high education levels (Table 3 and 6). Furthermore, we have conducted a lineal regression analysis, incorporating age and education level as covariables (Tables 5, 9, and 10). This analysis aims to account for potential biases introduced by the demographic composition of the sample (i.e. age and education).

It is essential to note that we recognize the limitations of our study, particularly the biases introduced by the overrepresentation of childless women and the age disparity between participants with and without children. We have explicitly described these limitations in the revised manuscript to provide transparency and context for our findings.

Now, Limitation reads (line 980):

… there was an overrepresentation of women (67.88%), individuals of younger age (averaging 21 years old), and those without children. This uneven representation introduces a potential bias in the data, underscoring the need for caution in interpreting the results. For future studies, it would be beneficial to aim for a sample more reflective of the Mexican population's demographics, where women constitute 51.2% and the median age is 29 years, as per INEGI's 2021 data [62]. Furthermore, the proportion of participants with children in our study was relatively small (15.77%). Although we assessed the interaction of gender and parental status in Study 3, the findings should be interpreted with care due to the aforementioned biases in sample representativeness regarding these variables.

Concerning the study's conceptual framework, there's a need to develop the concept of gender more comprehensively within the introduction, as explicitly addressed in the second paragraph. Mexico has a robust history of gender theory, including studies on masculinities. It is advisable to include references to the following:

* The concept of gender is multifaceted, historically contingent, and culturally shaped, and, therefore, is not static.

Response: We express gratitude for the guidance on developing the concept of gender within the introduction. The manuscript was revised to include references and provide a more comprehensive definition of gender, incorporating the multifaceted and culturally shaped nature of the concept in the context of the relationship between gender and parenthood (line 76):

Vespa [11] contends that gender ideology is dynamic, evolving over time as individuals encounter diverse social environments that introduce them to gender expectations related to aspects like marriage, parenthood, and work. Various determinants, such as race, age of children, education, family type, and pre-parenthood attitudes toward gender roles, contribute to this organic nature [11–13].

* One of the prevailing definitions of gender is the hegemonic model standard in Western societies, characterized by the sexual division of labor, which this study refers to as traditional gender roles. While the article references hegemonic gender roles, it lacks an explicit definition of sexual division of labor.

Response: We sincerely appreciate the suggestion to revisit the concept of the sexual division of labor, as it is a crucial aspect of social organization that profoundly influences family dynamics in Mexico. In the revised version of the introduction, you will find a clear and explicit definition of the concept (line 55):

In Western societies, gender ideology is molded by traditional gender roles defined by the sexual division of labor, where specific tasks are strictly assigned based on gender [1]. Women are typically tasked with reproductive work within the home and family, involving domestic management, infrastructure maintenance, and the care of family members. Conversely, men are expected to engage in productive work, contributing to the production of public goods and services, with economic retribution and social recognition [2].

It also helped us later in the General Discussion to better contextualize our findings, where we argue that this sexual division of labor could be one of the explanations for the disparities in the mental health outcomes regarding sex and parental status. You can find these insights at line 858:

The disparity in time allocation to unpaid work between men and women is still prominent, even in urban areas where women's professional and labor participation has grown. Women spend more than double the time on domestic tasks (such as food preparation, cleaning, caregiving, and voluntary work) compared to men [54]. Furthermore, as men increasingly engage in household chores and parenting [55], they may face discrimination for seeking work flexibility for childcare, along with ridicule and stigmatization from peers [14].

And line 895:

Women who step away from traditional motherhood roles to pursue employment often encounter mental health challenges due to the dual demands of professional and domestic responsibilities. Parenthood can potentially diminish the mental health benefits usually derived from other life experiences, such as career development, particularly for women with young children [56]. 

Moreover, it's essential to clarify that, in this article, the term "masculinity" pertains to the hegemonic model of masculinity. There are various models of masculinity, and this specificity should be underscored.

Response: We sincerely appreciate your comment regarding the different concepts of masculinity, in fact, the term we are referring to is traditional masculinity. You can read this clarification in the new version of the manuscript at line 61: 

This societal structure gives rise to traditional stereotypes of masculinity and femininity, representing the accepted beliefs about suitable characteristics for men and women [3]. Masculinity often involves traits like dominance, leadership, and competitiveness, while femininity is associated with qualities such as understanding, warmth, and compassion. These learned behaviors gradually become ingrained expectations, influencing individuals' actions and earning societal approval or disapproval [4], ultimately impacting their mental health and overall well-being.

In general, traditional masculinity traits have been associated with better physical health [5] and mental well-being. Specifically, traditional masculinity has shown a negative association with depression [6] and anxiety symptoms [7]. The influence of femininity traits on mental health remains inconclusive [8], as different studies have presented conflicting results. Some research suggests that femininity traits may increase the risk of depression in women [9], whereas other studies show a weak negative correlation between femininity and depression [6,7]. In addition, there is evidence of a positive correlation between femininity traits and anxiety symptoms [7]. Notably, women experience higher rates of physical and mental health problems than men in different age groups and regions around the world [10].

The introduction's fourth and fifth paragraphs discuss the challenges men and women face as they navigate the roles assigned by the sexual division of labor. It would be beneficial to provide context, highlighting that women have entered the workforce without a concurrent redistribution of responsibilities in the domestic sphere, which has implications for women's health.

Response: We provided an idea about women navigating through motherhood with a particular and more demanding context due to the sexual division of labor in the introduction section at line 99: 

For women, motherhood carries a unique context in which it is seen as both a gender-based obligation and a defining aspect of identity [20,21]. Moreover, motherhood, unlike fatherhood, is often described as intensive, meaning that it is seen as the most significant and valuable role a woman can assume. At the same time, it imposes strict demands and sets unattainable standards, such as the invisible double workload involving both productive and reproductive work [22, 2].

 However, we address the topic in a more elaborate way in the discussion section (line 855):

Even when parental roles conform to the gender equity model, there are situations that contribute to the maintenance of traditional parenthood dynamics based on gender. For instance, while women increasingly participate in the workforce, they continue to shoulder the bulk of reproductive work, leading to an often socially invisible double workload [2]. The disparity in time allocation to unpaid work between men and women is still prominent, even in urban areas where women's professional and labor participation has grown. Women spend more than double the time on domestic tasks (such as food preparation, cleaning, caregiving, and voluntary work) compared to men [54]. Furthermore, as men increasingly engage in household chores and parenting [55], they may face discrimination for seeking work flexibility for childcare, along with ridicule and stigmatization from peers [14].

From the introduction's 8th to 13th paragraph, methodological details are discussed, which would be more appropriate for the methods section rather than the introduction.

Response: Following your recommendation, we moved paragraphs 8 to 13 to the Method section of study 1 and 2. 

Now, Instruments and procedure of Study 1 reads (line 162):

We selected the ATGRS for our study as it is the sole scale developed in Mexico specifically designed to evaluate the construct it measures: an individual's perception based on societal norms and expectations related to traditional and non-traditional gender roles in men and women [33]. The authors conducted an exploratory study prior to developing the scale to ensure its content was both culturally sensitive and had face validity. They used open questions to collect diverse conceptions -encompassing behaviors, traits, beliefs, etc.- that men and women have regarding their gender identity [33]. The scale's validation process involved 120 men and 224 women, with an average age of 30 years (SD = 9.91). The educational background of the participants was varied: 40% had an education level of high school or less, while 60% had attained at least a bachelor's degree. All participants belonged to a medium socioeconomic stratum.

 ATGRS comprises 21 items rated on a 5-point Likert scale (1= I like it very much, 5=I dislike it very much). The scale is structured around three factors, collectively accounting for 46% of the variance. The first factor, Traditional Attitudes Towards Gender Roles, includes statements that endorse the continuation of conventional gender roles. The second factor, Favorable Attitudes Towards Gender Equity, reflects a supportive view of equal rights and opportunities for both men and women. The third factor, Favorable Attitudes Towards Female Empowerment, consists of items that positively assess the progress and emancipation of women. The Cronbach’s alpha coefficients, which measure internal consistency, were high, indicating strong reliability: 0.86 for the first factor, 0.85 for the second, and 0.76 for the third.

 Our analysis identified several issues with the composition of the factors within the ATGRS. Firstly, the scale includes items that are anchored in hegemonic gender roles. Even the second factor, which is intended to assess favorable attitudes towards gender equity, does so within a traditional framework. For instance, the item "That the man participates in the care of the children" reflects the conventional view that caregiving is primarily a woman's role. Thus, within the scale, male involvement in childcare is seen as an equitable attitude. However, in a modern context, true equity in childcare would mean recognizing both parents' rights and responsibilities in child-rearing. This suggests a need for additional items in this factor to more accurately encapsulate the essence of gender equity.

 Another issue is with the third factor, Favorable Attitude Towards Female Empowerment. Most of its items seem to focus more on gender equity rather than on female empowerment. Take, for example, "That women have the same freedom as men" (Item 3 in the original scale) and "That men and women perform the same tasks" (Item 6 in the original scale). Considering empowerment as a process enabling women, particularly those who have faced oppression, to make independent and strategic life choices based on their personal priorities [35], it becomes evident that the items in this third factor fall short of adequately capturing the concept of female empowerment.

 To address these issues, we modified the items that make up factor 1 of the original scale, Favorable attitude towards traditional roles, but with reversed roles. For instance, the original item "That the man establishes the rules of the home" was complemented with "That the woman establishes the rules of the home." In the second factor, Favorable Attitude Towards Gender Equity, we introduced role-reversed items such as "That the woman expresses her emotions just like the man" and "That the man focuses on personal and professional self-improvement." Additionally, we incorporated two items related to maternity/paternity to address specific gender role stereotypes associated with women [21]. These items were "That the success of being a woman lies in being a mother" and "That the success of being a man lies in being a father." This inclusion acknowledges the gendered expectations surrounding parenting roles. In total, 13 new items were added to the scale, bringing the total number of items in the extended version of the ATGRS to 34. All participants in our study responded to this expanded scale via a Google Form. 

And, Instruments and procedure of Study 2 reads (line 337):

In Mexico there is just one validated scale that evaluates Motherhood Beliefs, i.e. MBS [34]. This scale was developed through interviews with three distinct groups of women: 1) those receiving gynecological care at a reproductive health hospital, 2) mothers of preschool children, and 3) university students. These women were asked about their perceptions of what it means to be a mother. Based on their responses, 17 items were written with Likert-type responses ranging from 0 = disagree to 5 = totally agree belonging to two dimensions: 1) Sense of Life, which assesses statements related to motherhood as a woman's life purpose, something that brings her happy and fulfilled; and 2) Social Duty, which includes statements about motherhood as an obligation to society, with a woman's value being diminished if she does not fulfill this duty. After initial development, three judges assessed how well each item aligned with the definitions of these proposed dimensions, leading to the elimination of two items.

 The scale's validation involved a sample of 545 women, aged between 17 and 52 years (Mean=30.97; SD=7.54). To establish concurrent validity, the study included women with varying maternal statuses: those who experienced perinatal loss (35.8%), those undergoing infertility treatments or with high-risk pregnancies (10.8%), mothers of preschool-age children (36.6%), and female university students without children (17.1%). Educational background varied, with 64.7% having at least a secondary school education. Only 38.2% were employed, and about 40% were single. Approximately 40% of the participants had between one and five children.

 Two additional items were excluded due to their factor loadings being less than 0.40 in the exploratory analysis, resulting in the final scale comprising 13 items. Again, the scale was categorized into two factors: 1) Sense of Life, and 2) 'Social Duty. The scale demonstrated satisfactory internal consistency, with a Cronbach's alpha of 0.92 for the total scale, 0.91 for the Sense of Life factor, and 0.83 for the Social Duty factor.

 During the validation of the Motherhood Belief Scale, groups were contrasted based on maternal status. Findings revealed that women who had experienced perinatal loss or infertility issues exhibited more favorable beliefs towards motherhood on both subscales than did mothers and university students.

 Although the initial validation included women with diverse motherhood statuses, we consider that the 13 items across just two factors are insufficient for a comprehensive assessment of motherhood in our society [37]. The original scale's items also displayed a significant bias towards hegemonic gender stereotypes, exemplified by statements like "A woman is complete until she is a mother". 

 To enhance the scale, we introduced additional items that more thoroughly examine various aspects of motherhood while avoiding gender stereotypes. These new items were derived from an extensive review of theoretical and scientific literature on motherhood from 2000 to 2021. This review led to the identification of a wide range of definitions, concepts, and beliefs about motherhood, which were then adapted into Likert-type statements. As a result, 16 new items were added to the MBS, covering areas such as female identity, decision-making, lifestyle, success, life plans, and emotions including love, frustration, pain, and passion. For example, new items included 'Maternity is a woman's decision' and 'Motherhood can generate anxiety,' as detailed in Table 2. Participants completed this extended version of the MBS via a Google Form.

Lastly, in the final paragraph of the introduction, there is a notable absence of a reference to the gaps this study aims to address concerning the relationships between gender roles, conceptions of motherhood, and mental health within the study population.

Response: Thank you for your comments. We sincerely believe that the introduction will be better interpreted after your suggestions. In this new version of the introduction, we have added the requested information. 

First, we added information referring to the study population (line 109):

In Mexico, family and gender roles adhere strictly to societal expectations, imposing significant pressure to conform. A central cultural aspect is affiliative obedience, where both men and women embody the distinct feature of Mexican culture: self-sacrifice. This implies a belief that prioritizing the needs of others over one's own is more important [28]. Mexicans frequently prioritize external social demands over their personal interests and desires [29].

The traditional Mexican family serves as an illustrative context to understand how gender roles are instilled and acquired in Mexico. According to Diaz-Loving et al. [28], Mexican families are guided by two main axes: a) the power and absolute supremacy of the father, and b) the love and essential sacrifice of the mother. This family structure is mirrored in normative traits of Mexicans, including the emphasis on obedience, the fear of family dishonor, and the magnification of virginity even among adolescents. Moreover, predominant beliefs in this culture include machismo, fear of parental authority, and the importance of respect.

Nevertheless, in Mexico there is a growing diversification of family structures, contemporary meanings related to the family are going beyond the exclusivity of the nuclear biparental model and there is greater openness and tolerance regarding possible alternative forms of coexistence and personal and family development [30]. Within this diversity, coupled with the collectivist character of Mexican society [31], is the participation of networks of women and extended family in the care of children, a term known as othermothers [32]. This could facilitate some conditions such as women having a support network that allows them to occupy other professional and work roles or to have time for self-care, reducing the parental burden.

And later we argue the importance of addressing the relationship between gender roles and motherhood beliefs and mental health specifically in the study population (line 129):

Despite current changes in parenting roles and parenting decisions, it seems that there is an underlying maintenance of traditional gender roles, and if so, it would be important to know their relationship with mental health in both people with and without children. Furthermore, there is a lack of research on what this relationship would be like in a society where traditional family and gender beliefs are highly esteemed, and where women, due to the conventional family structure, are engaged in childcare collectively rather than individually. To test this hypothesis, the first step would be to have updated instruments that probe attitudes towards gender roles today, otherwise the results between gender roles and mental health could be biased.

As for the methods section, in addition to the previously mentioned general comments, the following is requested:

* To include more extensive information about the participants, such as their geographical diversity within Mexico.

Response: Thank you for your comment, in response to your request we have added demographic information of the participants such as their last grade of studies, occupation, locality and marital status. 

In the Study 3 participants section, you can read (line 505):

Participants were native-Spanish speakers of Mexican nationality, and all signed a written privacy notice and an informed consent form. Demographic data were anonymously collected including age, last grade of studies, occupation, locality, and marital status. 

You can find all demographic data in new Table 3 (line 586):

 PARENTING STATUS

 Without children With children

 n=577 n=108

 SEX SEX

 Women Men Non-binary Women Men Non-binary

 n=373 n=192 n=12 n=92 n=15 n=1

 Mean (sd) Mean (sd) Mean (sd) Mean (sd) Mean (sd) Mean (sd)

Age 21.23 (4.04) 22.64 (6.30) 20.08 (1.51) 40.04 (10.01) 47.27 (11.00) 45 (--)

Last grade of studies n (%) n (%) n (%) n (%) n (%) n (%)

none 0 0 0 1 (1.09) 0 0

primary school 0 0 0 3 (3.26) 0 0

secondary school 10 (2.62) 5 (2.60) 0 14 (15.22) 3 (20.00) 0

high school 263 (70.51) 117 (60.94) 9 (75) 26 (28.26) 3 (20.00) 0

bachelor’s degree 92 (24.66) 66 (34.38) 3 (25) 44 (47.83) 6 (40.00) 0

postgraduate degree 8 (2.14) 4 (2.08) 0 4 (4.35) 3 (20.00) 1 (100)

Occupation n (%) n (%) n (%) n (%) n (%) n (%)

Student 318 (85.25) 131 (68.23) 10 (83.33) 2 (2.17) 0 0

Part-time job 17 (4.56) 16 (8.33) 2 (16.67) 17 (18.48) 2 (13.33) 0

Full-time job 23 (6.17) 35 (18.23) 0 37 (40.22) 8 (53.33) 1 (100)

Unpaid job 12 (3.22) 3 (1.56) 0 29 (31.52) 1 (6.67) 0

Other 3 (0.80) 7 (3.65) 0 7 (7.61) 4 (26.67) 0

Locality n (%) n (%) n (%) n (%) n (%) n (%)

Mexico City 279 (74.80) 146 (76.04) 8 (66.67) 65 (70.65) 10 (66.67) 0

Mexico state 77 (20.64) 41 (21.35) 3 (25.00) 21 (22.83) 3 (20) 1 (100)

Other state 17 (4.56) 5 (2.60) 1 (8.33) 6 (6.52) 2 (13.33) 0

Marital Status n (%) n (%) n (%) n (%) n (%) n (%)

Single 359 (96.25) 174 (90.63) 12 (100) 38 (41.30) 5 (33.33) 1 (100)

Married 12 (3.22) 15 (7.81) 0 54 (58.70) 10 (66.67) 0

Engagement 2 (0.54) 3 (1.56) 0 0 0 0

* To include a synthesis of studies 1 and 2. The synthesis should demonstrate that the scales are appropriate for measuring what they intend to measure and valid for the Mexican context. If the scales were validated in populations with specific characteristics (e.g., more women, childless, educated, etc.), whether they are valid for the broader Mexican population should be clarified.

Response: Thank you for this valuable suggestion. We consider that the section corresponding to measuring instruments will be more complete by adding your recommendations. We add the information required in Study 1 to the line 161:

Instruments and procedure

We selected the ATGRS for our study as it is the sole scale developed in Mexico specifically designed to evaluate the construct it measures: an individual's perception based on societal norms and expectations related to traditional and non-traditional gender roles in men and women [33]. The authors conducted an exploratory study prior to developing the scale to ensure its content was both culturally sensitive and had face validity. They used open questions to collect diverse conceptions -encompassing behaviors, traits, beliefs, etc.- that men and women have regarding their gender identity [33]. The scale's validation process involved 120 men and 224 women, with an average age of 30 years (SD = 9.91). The educational background of the participants was varied: 40% had an education level of high school or less, while 60% had attained at least a bachelor's degree. All participants belonged to a medium socioeconomic stratum.

ATGRS comprises 21 items rated on a 5-point Likert scale (1= I like it very much, 5=I dislike it very much). The scale is structured around three factors, collectively accounting for 46% of the variance. The first factor, Traditional Attitudes Towards Gender Roles, includes statements that endorse the continuation of conventional gender roles. The second factor, Favorable Attitudes Towards Gender Equity, reflects a supportive view of equal rights and opportunities for both men and women. The third factor, Favorable Attitudes Towards Female Empowerment, consists of items that positively assess the progress and emancipation of women. The Cronbach’s alpha coefficients, which measure internal consistency, were high, indicating strong reliability: 0.86 for the first factor, 0.85 for the second, and 0.76 for the third.

And in study 2, in the line 336:

Instruments and procedure

In Mexico there is just one validated scale that evaluates Motherhood Beliefs, i.e. MBS [34]. This scale was developed through interviews with three distinct groups of women: 1) those receiving gynecological care at a reproductive health hospital, 2) mothers of preschool children, and 3) university students. These women were asked about their perceptions of what it means to be a mother. Based on their responses, 17 items were written with Likert-type responses ranging from 0 = disagree to 5 = totally agree belonging to two dimensions: 1) Sense of Life, which assesses statements related to motherhood as a woman's life purpose, something that brings her happy and fulfilled; and 2) Social Duty, which includes statements about motherhood as an obligation to society, with a woman's value being diminished if she does not fulfill this duty. After initial development, three judges assessed how well each item aligned with the definitions of these proposed dimensions, leading to the elimination of two items.

The scale's validation involved a sample of 545 women, aged between 17 and 52 years (Mean=30.97; SD=7.54). To establish concurrent validity, the study included women with varying maternal statuses: those who experienced perinatal loss (35.8%), those undergoing infertility treatments or with high-risk pregnancies (10.8%), mothers of preschool-age children (36.6%), and female university students without children (17.1%). Educational background varied, with 64.7% having at least a secondary school education. Only 38.2% were employed, and about 40% were single. Approximately 40% of the participants had between one and five children.

Two additional items were excluded due to their factor loadings being less than 0.40 in the exploratory analysis, resulting in the final scale comprising 13 items. Again, the scale was categorized into two factors: 1) Sense of Life, and 2) 'Social Duty. The scale demonstrated satisfactory internal consistency, with a Cronbach's alpha of 0.92 for the total scale, 0.91 for the Sense of Life factor, and 0.83 for the Social Duty factor.

During the validation of the Motherhood Belief Scale, groups were contrasted based on maternal status. Findings revealed that women who had experienced perinatal loss or infertility issues exhibited more favorable beliefs towards motherhood on both subscales than did mothers and university students.

Although the initial validation included women with diverse motherhood statuses, we consider that the 13 items across just two factors are insufficient for a comprehensive assessment of motherhood in our society [37]. The original scale's items also displayed a significant bias towards hegemonic gender stereotypes, exemplified by statements like "A woman is complete until she is a mother".

To enhance the scale, we introduced additional items that more thoroughly examine various aspects of motherhood while avoiding gender stereotypes. These new items were derived from an extensive review of theoretical and scientific literature on motherhood from 2000 to 2021. This review led to the identification of a wide range of definitions, concepts, and beliefs about motherhood, which were then adapted into Likert-type statements. As a result, 16 new items were added to the MBS, covering areas such as female identity, decision-making, lifestyle, success, life plans, and emotions including love, frustration, pain, and passion. For example, new items included 'Maternity is a woman's decision' and 'Motherhood can generate anxiety,' as detailed in Table 2. Participants completed this extended version of the MBS via a Google Form.

* Explain the study's independent, covariate, and outcome variables, and detail the instruments used to measure them. While the instruments used are listed, it is crucial to establish what constructs these scales are measuring, their origins, and whether they have been validated for use in Mexico.

Response: Following your recommendation, we added information about the study's independent, covariate, and outcome variables. You can read this at line 538:

Data analysis 

Initially, we calculated descriptive statistics for the sociodemographic variables, including measures of central tendency (such as mean and standard deviation or median and interquartile range) and the cumulative percentage frequency of last grade of studies, occupation, locality and marital status. As was expected, people with children were found to be older and had higher education levels compared to those without children. Consequently, age and education level were incorporated as covariates in association analyses to control for these demographic differences. 

Next, we conducted partial correlation analyses using Spearman's non-parametric method to examine the relationships between the different scales while controlling for the effects of covariates, namely age and education level. To account for multiple correlations and reduce the risk of Type I errors, we applied a Bonferroni correction. Additionally, as a complement analysis, using the Ordinary Least Squares (OLS) method, we conducted multiple linear regressions to predict a dependent variable from a set of factors on a scale. For instance, to predict scores on the Social Duty factor of the MBS scale, all factors from the ATGRS scale were used as predictor variables. In all cases, the interaction with sex and parental status was calculated, and adjustments were made for age and educational level.

Moreover, we extended the information about the instruments, for Study 1 you can read (line 161):

Instruments and procedure

We selected the ATGRS for our study as it is the sole scale developed in Mexico specifically designed to evaluate the construct it measures: an individual's perception based on societal norms and expectations related to traditional and non-traditional gender roles in men and women [33]. The authors conducted an exploratory study prior to developing the scale to ensure its content was both culturally sensitive and had face validity. They used open questions to collect diverse conceptions -encompassing behaviors, traits, beliefs, etc.- that men and women have regarding their gender identity [33]. The scale's validation process involved 120 men and 224 women, with an average age of 30 years (SD = 9.91). The educational background of the participants was varied: 40% had an education level of high school or less, while 60% had attained at least a bachelor's degree. All participants belonged to a medium socioeconomic stratum.

ATGRS comprises 21 items rated on a 5-point Likert scale (1= I like it very much, 5=I dislike it very much). The scale is structured around three factors, collectively accounting for 46% of the variance. The first factor, Traditional Attitudes Towards Gender Roles, includes statements that endorse the continuation of conventional gender roles. The second factor, Favorable Attitudes Towards Gender Equity, reflects a supportive view of equal rights and opportunities for both men and women. The third factor, Favorable Attitudes Towards Female Empowerment, consists of items that positively assess the progress and emancipation of women. The Cronbach’s alpha coefficients, which measure internal consistency, were high, indicating strong reliability: 0.86 for the first factor, 0.85 for the second, and 0.76 for the third.

For Study 2, we added the requested information at line 336:

Instruments and procedure

In Mexico there is just one validated scale that evaluates Motherhood Beliefs, i.e. MBS [34]. This scale was developed through interviews with three distinct groups of women: 1) those receiving gynecological care at a reproductive health hospital, 2) mothers of preschool children, and 3) university students. These women were asked about their perceptions of what it means to be a mother. Based on their responses, 17 items were written with Likert-type responses ranging from 0 = disagree to 5 = totally agree belonging to two dimensions: 1) Sense of Life, which assesses statements related to motherhood as a woman's life purpose, something that brings her happy and fulfilled; and 2) Social Duty, which includes statements about motherhood as an obligation to society, with a woman's value being diminished if she does not fulfill this duty. After initial development, three judges assessed how well each item aligned with the definitions of these proposed dimensions, leading to the elimination of two items.

The scale's validation involved a sample of 545 women, aged between 17 and 52 years (Mean=30.97; SD=7.54). To establish concurrent validity, the study included women with varying maternal statuses: those who experienced perinatal loss (35.8%), those undergoing infertility treatments or with high-risk pregnancies (10.8%), mothers of preschool-age children (36.6%), and female university students without children (17.1%). Educational background varied, with 64.7% having at least a secondary school education. Only 38.2% were employed, and about 40% were single. Approximately 40% of the participants had between one and five children.

Two additional items were excluded due to their factor loadings being less than 0.40 in the exploratory analysis, resulting in the final scale comprising 13 items. Again, the scale was categorized into two factors: 1) Sense of Life, and 2) 'Social Duty. The scale demonstrated satisfactory internal consistency, with a Cronbach's alpha of 0.92 for the total scale, 0.91 for the Sense of Life factor, and 0.83 for the Social Duty factor.

During the validation of the Motherhood Belief Scale, groups were contrasted based on maternal status. Findings revealed that women who had experienced perinatal loss or infertility issues exhibited more favorable beliefs towards motherhood on both subscales than did mothers and university students.

Although the initial validation included women with diverse motherhood statuses, we consider that the 13 items across just two factors are insufficient for a comprehensive assessment of motherhood in our society [37]. The original scale's items also displayed a significant bias towards hegemonic gender stereotypes, exemplified by statements like "A woman is complete until she is a mother".

To enhance the scale, we introduced additional items that more thoroughly examine various aspects of motherhood while avoiding gender stereotypes. These new items were derived from an extensive review of theoretical and scientific literature on motherhood from 2000 to 2021. This review led to the identification of a wide range of definitions, concepts, and beliefs about motherhood, which were then adapted into Likert-type statements. As a result, 16 new items were added to the MBS, covering areas such as female identity, decision-making, lifestyle, success, life plans, and emotions including love, frustration, pain, and passion. For example, new items included 'Maternity is a woman's decision' and 'Motherhood can generate anxiety,' as detailed in Table 2. Participants completed this extended version of the MBS via a Google Form.

* Link each analysis to its specific objective.

Response: Thanks for the suggestion. In the Data Analysis section of Study 3, we added its corresponding statistical analysis to each specific objective, you can read it at line 546:

To achieve the primary aim of the Study 3, which is to examine the association between the new versions of ATGRS and MBS, as well as their correlation with measures of mental health, we conducted partial correlation analyses using Spearman's non-parametric method to examine the relationships between the different scales while controlling for the effects of covariates, namely age and education level. To account for multiple correlations and reduce the risk of Type I errors, we applied a Bonferroni correction. Additionally, as a complement analysis, using the Ordinary Least Squares (OLS) method, we conducted multiple linear regressions to predict a dependent variable from a set of factors on a scale. For instance, to predict scores on the Social Duty factor of the MBS scale, all factors from the ATGRS scale were used as predictor variables. In all cases, the interaction with sex and parental status was calculated, and adjustments were made for age and educational level.

 In addition to the main objective, we aimed to assess these variables concerning sex and parental status: ATGRS and MBS in Study 3.2 and measures of mental health in Study 3.4. To achieve this, we utilized the Mann-Whitney test for independent samples to compare the scores of all scales based on parenting status and sex. To mitigate the possibility of Type I errors resulting from multiple comparisons, we applied the Holm correction. All statistical analyses were performed using R Studio software, version 4.1.1 [36], with a significance level set at p < 0.05.

Within the results section:

* Include a descriptive table presenting the overall results of the measurement instruments used in the sample.

Response: You can find this information at the bottom of new Table 3 (line 586)

 PARENTING STATUS

 Without children With children

 n=577 n=108

 SEX SEX

 Women Men Non-binary Women Men Non-binary

 n=373 n=192 n=12 n=92 n=15 n=1

 Mean (sd) Mean (sd) Mean (sd) Mean (sd) Mean (sd) Mean (sd)

Age 21.23 (4.04) 22.64 (6.30) 20.08 (1.51) 40.04 (10.01) 47.27 (11.00) 45 (--)

Last grade of studies n (%) n (%) n (%) n (%) n (%) n (%)

none 0 0 0 1 (1.09) 0 0

primary school 0 0 0 3 (3.26) 0 0

secondary school 10 (2.62) 5 (2.60) 0 14 (15.22) 3 (20.00) 0

high school 263 (70.51) 117 (60.94) 9 (75) 26 (28.26) 3 (20.00) 0

bachelor’s degree 92 (24.66) 66 (34.38) 3 (25) 44 (47.83) 6 (40.00) 0

postgraduate degree 8 (2.14) 4 (2.08) 0 4 (4.35) 3 (20.00) 1 (100)

Occupation n (%) n (%) n (%) n (%) n (%) n (%)

Student 318 (85.25) 131 (68.23) 10 (83.33) 2 (2.17) 0 0

Part-time job 17 (4.56) 16 (8.33) 2 (16.67) 17 (18.48) 2 (13.33) 0

Full-time job 23 (6.17) 35 (18.23) 0 37 (40.22) 8 (53.33) 1 (100)

Unpaid job 12 (3.22) 3 (1.56) 0 29 (31.52) 1 (6.67) 0

Other 3 (0.80) 7 (3.65) 0 7 (7.61) 4 (26.67) 0

Locality n (%) n (%) n (%) n (%) n (%) n (%)

Mexico City 279 (74.80) 146 (76.04) 8 (66.67) 65 (70.65) 10 (66.67) 0

Mexico state 77 (20.64) 41 (21.35) 3 (25.00) 21 (22.83) 3 (20) 1 (100)

Other state 17 (4.56) 5 (2.60) 1 (8.33) 6 (6.52) 2 (13.33) 0

Marital Status n (%) n (%) n (%) n (%) n (%) n (%)

Single 359 (96.25) 174 (90.63) 12 (100) 38 (41.30) 5 (33.33) 1 (100)

Married 12 (3.22) 15 (7.81) 0 54 (58.70) 10 (66.67) 0

Engagement 2 (0.54) 3 (1.56) 0 0 0 0

 Mean (sd) Mean (sd) Mean (sd) Mean (sd) Mean (sd) Mean (sd)

Depression 19.35 (12.48) 13.95 (11.29) 32.83 (15.42) 12.91 (11.56) 8.93 (8.94) 4

Anxiety 19.13 (13.26) 12.43 (12.27) 26.92 (14.85) 12.72 (12.36) 9.47 (9.83) 1

PPF 133.14 (29.48) 142.79 (32.21) 114.58 (26.09) 141.14 (36.44) 155.93 (25.54) 130

ATGRS Mean (sd) Mean (sd) Mean (sd) Mean (sd) Mean (sd) Mean (sd)

Traditional 15.96 (4.61) 19.78 (5.08) 17.42 (3.40) 16.83 (5.36) 19.87 (4.55) 12

Egalitarian success 57.42 (4.71) 53.83 (9.05) 57.33 (3.23) 55.34 (5.99) 50.53 (11.04) 49

Gender equity 8.00 (3.01) 8.58 (3.42) 5.92 (2.07) 9.22 (3.54) 11.07 (3.37) 6

Non-traditional 19.51 (5.48) 19.23 (4.70) 20.00 (2.92) 17.52 (5.34) 17.6 (3.78) 26

MBS Mean (sd) Mean (sd) Mean (sd) Mean (sd) Mean (sd) Mean (sd)

Social Duty 0.26 (1.24) 0.34 (1.49) 0.08 (0.29) 0.63 (1.94) 1.13 (3.02) 0

Sense of life 2.17 (3.91) 4.32 (5.68) 1.50 (1.57) 7.71 (8.65) 5.87 (7.8) 0

Hegemonic stereotypes 9.23 (4.67) 9.43 (5.34) 11.17 (5.83) 9.52 (5.69) 9.13 (5.84) 13

Negative emotions 7.83 (2.33) 7.23 (2.49) 6.92 (2.27) 6.61 (3.28) 5.6 (3.27) 10

Motherhood as a decision 36.62 (5.60) 33.96 (8.19) 37.67 (2.06) 33.63 (6.84) 30.53 (8.67) 40

* Regarding the first objective (examining the relationship between the Attitudes Towards Gender Roles Scale and the Motherhood Beliefs Scale, stratified by parental status), the results of the correlations between the scales are described. However, a comparison of the dimensions of these correlations needs to be included. Are specific correlations more significant than others?

Response: Thank you for your insightful comment. We acknowledge the importance of providing a comprehensive analysis of the correlations between the Attitudes Towards Gender Roles Scale (ATGRS) and the Motherhood Beliefs Scale (MBS) adjusted by parental status. In our updating revision, we have included multiple lineal models adjusted by age and education, and we explored the interaction with sex and parental status. This additional analysis enhances the clarity and depth of our findings, allowing for a more nuanced understanding of the association between gender roles and motherhood beliefs. Your suggestion is valuable in presenting a more robust interpretation of our results. At line 617:

To confirm and complement these findings, we employed a multiple linear regression model to explore how all factors from the ATGRS scale influence each component of the MBS (Table 5). Moreover, we investigated the interaction of these factors with sex and parenthood status, while controlling for age and educational level. Regarding Social duty, the model demonstrated an adjusted R2 value of 0.11 (F(13, 671) = 7.71, p < 0.001). For the Sense of life, the model exhibited an adjusted R2 value of 0.21 (F(13, 671) = 15.48, p < 0.001). Concerning Hegemonic stereotypes, the model indicated an adjusted R2 value of 0.06 (F(13, 671) = 4.46, p < 0.001). For Negative emotions, the model displayed an adjusted R2 value of 0.07 (F(13, 671) = 5.39, p < 0.001). Regarding Motherhood as a decision, the model revealed an adjusted R2 value of 0.17 (F(13, 671) = 12.29, p < 0.001). No interactions with sex and parenthood status were detected in any model.

Table 5. Multiple lineal model from ATGRS factors for each factor of MBS, adjusted for age and educational level. 

 Estimate SE t-value p-value

Social duty Traditional 0.05 0.01 3.67 p < 0.001

 Egalitarian success 0.05 0.02 2.71 p < 0.01

 Gender equity -0.05 0.01 -5.95 p < 0.001

 Non-traditional -0.05 0.02 -2.11 p < 0.05

Sense of life Traditional 0.24 0.05 4.53 p < 0.001

 Egalitarian success 0.41 0.07 6.12 p < 0.001

 Gender equity -0.04 0.03 -1.33 N.S

 Non-traditional -0.15 0.08 -1.72 N.S

Hegemonic stereotypes Traditional 0.151 0.05 2.88 p < 0.01

 Egalitarian success 0.194 0.06 2.98 p < 0.01

 Gender equity -0.04 0.03 -1.29 N.S

 Non-traditional 0.01 0.08 0.074 N.S

Negative emotions Traditional -0.02 0.02 -0.85 N.S

 Egalitarian success -0.06 0.03 -1.77 N.S

 Gender equity 0.06 0.02 4.02 p < 0.001

 Non-traditional 0.09 0.04 1.98 p < 0.05

Motherhood as a decision Traditional -0.06 0.06 -1.05 N.S

 Egalitarian success -0.39 0.08 -4.84 p < 0.001

 Gender equity 0.28 0.04 7.25 p < 0.001

 Non-traditional 0.13 0.11 1.26 N.S

N.S., not significant

The results of the multiple linear regression are also in accordance with the hypotheses. Social duty, Sense of life, and Hegemonic stereotypes as motherhood beliefs were predicted positively by traditional factors of ATGRS (Table 5). Non-traditional factors of ATGRS predicted negatively Social duty, while a contrasting pattern was detected for the Negative emotions factor of the MBS scale, i.e., it was predicted positively. Motherhood as a decision was predicted positively by Gender equity and negatively by Egalitarian success.

* For the second objective (examining differences in ATGRS and MBS based on sex and parental status), it is noted that the sample was adjusted without controlling for age and education levels in people with and without children, as they were not the primary focus of the study. This explanation is considered insufficient. It is requested to include a clarification regarding this issue, especially regarding the low representation of men in the sample.

Response: Thank you for your feedback. We appreciate your attention to detail. In response to your concern, we have revised our approach. In this new version, we provide more detailed information about the demographic characteristics of the participants. Since older individuals tend to have higher education levels and are more likely to be parents than younger ones, it has been challenging to disentangle the effects of these variables. Although for the analyses of the association between variables we controlled age and educational level, in the comparison analyses it was not possible due to the characteristics of the subsamples. However, we explicitly include results regarding these differences to add more transparency to our findings. We consider that it is necessary to address this issue of analyzing the individual effects of the age and education variables in future studies.

At line 541:

As was expected, people with children were found to be older and had higher education levels compared to those without children. Consequently, age and education level were incorporated as covariates in association analyses to control for these demographic differences. 

At line 658:

In demographic data, we detected differences in age and education level between these subgroups; people with children were older than the group without children (W = 10727, p-value < 0.001), and people without children had more years of study, in proportion, to those with children (X-squared = 40.58, df = 5, p-value < 0.001, Table 6).

Additionally, we explicitly address as limitations the issue of the low representation of men in the sample, offering insights into how this may influence the generalizability of our findings. Your input has contributed to a more thorough and transparent presentation of our methodology.

At line 980: 

… there was an overrepresentation of women (67.88%), individuals of younger age (averaging 21 years old), and those without children. This uneven representation introduces a potential bias in the data, underscoring the need for caution in interpreting the results. For future studies, it would be beneficial to aim for a sample more reflective of the Mexican population's demographics, where women constitute 51.2% and the median age is 29 years, as per INEGI's 2021 data [62]. Furthermore, the proportion of participants with children in our study was relatively small (15.77%). Although we assessed the interaction of gender and parental status in Study 3, the findings should be interpreted with care due to the aforementioned biases in sample representativeness regarding these variables. Further research is warranted to delve into the influence of social constructs on gender roles and motherhood, utilizing larger and more diverse samples of men and women with children. 

* The research's central focus is the third objective, which examines the relationship between ATGRS and MBS with mental health. However, the absence of a gender-based comparison and its implications for the results still need to be addressed.

Response: We appreciate your valuable feedback and agree that a gender-based comparison is crucial to enhance the interpretation of our results. In the revised version, we have incorporated a multiple lineal model analysis to explore potential interaction in the relationship between ATGRS, MBS, and mental health with sex and parental status, which can be found at tables 9 and 10 (line 723): 

To corroborate our findings, we employed a multiple linear regression model to analyze the impact of factors from both the ATGRS and the MBS on mental health measures. This analysis also considered the interaction of these factors with sex and parental status, while adjusting for age and educational level. The model that used ATGRS factors to predict depression yielded an adjusted R2 value of 0.11 (F(13, 671) = 5.63, p < 0.001). In the case of anxiety, the model demonstrated an adjusted R2 value of 0.07 (F(13, 671) = 5.11, p < 0.001). Notably, there was a significant sex-related effect, with women showing higher anxiety scores (estimates = 10.12, SE = 4.69, t = 1.15, p < 0.05). Regarding PPF, the model revealed an adjusted R2 value of 0.08 (F(13, 671) = 5.09, p < 0.001). Traditional factors negatively predicted both depression and anxiety. Conversely, Non-traditional factor were positively associated with depression and Gender equity was negatively associated with PPF (Table 9).

Table 9. Multiple lineal model from ATGRS factors for depression, anxiety and PPF, adjusted for age and educational level. 

 Estimate SE t-value p-value

Depression Traditional -0.28 0.13 -2.15 p < 0.05

 Egalitarian success 0.03 0.16 0.17 N.S.

 Gender equity -0.01 0.07 -0.18 N.S.

 Non-traditional 0.42 0.21 1.97 p < 0.05

Anxiety Traditional -0.38 0.14 -2.74 p < 0.01

 Egalitarian success 0.05 0.17 0.31 N.S.

 Gender equity -0.10 0.08 -1.12 N.S.

 Non-traditional 0.05 0.02 1. N.S.

PPF Traditional 0.36 0.32 1.09 N.S.

 Egalitarian success -0.39 0.40 -0.96 N.S.

 Gender equity 0.93 0.18 4.94 p < 0.001

 Non-traditional -0.89 0.54 -1.64 N.S.

N.S., not significant

In the multiple linear regression model where MBS factors served as predictors for depression, an adjusted R2 value of 0.11 was observed (F(14, 670) = 7.06, p < 0.001). Regarding anxiety, the model showed an adjusted R2 value of 0.07 (F(14, 670) = 4.73, p < 0.001). For PPF, the adjusted R2 value was 0.04 (F(14, 670) = 3.51, p < 0.001). Notably, in these models, no interactions with sex and parental status were evident. Expanding upon the results of the correlation analysis, Negative emotions, as a belief related to motherhood, positively predicts both depression and anxiety. However, it had a negative predictive relationship with PPF (Table 10).

Table 10. Multiple lineal model from MBS factors for depression, anxiety and PPF, adjusted for age and educational level. 

 Estimate SE t-value p-value

Depression Social Duty 0.58 0.36 1.58 N.S.

 Sense of life -0.17 0.11 -1.64 N.S.

 Hegemonic stereotypes 0.03 0.11 0.33 N.S.

 Negative emotions 0.85 0.19 4.43 p < 0.001

 Motherhood as a decision -0.06 0.11 -0.63 N.S.

Anxiety Social Duty 0.41 0.40 1.04 N.S.

 Sense of life -0.11 0.11 -1.03 N.S.

 Hegemonic stereotypes 0.01 0.11 0.16 N.S.

 Negative emotions 0.41 0.21 1.96 p < 0.05

 Motherhood as a decision -0.13 0.11 -1.21 N.S.

PPF Social Duty -0.58 0.96 -0.61 N.S.

 Sense of life 0.01 0.27 0.06 N.S.

 Hegemonic stereotypes 0.42 0.26 1.61 N.S.

 Negative emotions -1.41 0.50 -2.79 p < 0.01

 Motherhood as a decision 0.70 0.27 2.52 p < 0.05

N.S., not significant

This addition contributes to a more comprehensive understanding of how sex interacts with the dynamics between gender roles, motherhood beliefs, and mental health outcomes. Thank you for pointing out this important aspect, and we are committed to addressing it in our revised manuscript.

In the general discussion, it is critical to recognize that the research methodology employed does not enable the establishment of causal relationships, for instance, about the idea that having a child primarily had a 'traditionalizing' effect. Even if the results show that childless people are more likely to hold non-hegemonic beliefs while individuals with children obtained higher scores in traditional attitudes about gender roles, it is unclear if there is a reverse causality. It could be that people with children have always been more conservative and, consequently, have children. Alternatively, generational variables may confound the results, given that the childless group is around 20 years old, while the group with children is around 40. Finally, in the last sentence of the discussion, it is possible to state that the study shows a correlation between these variables. Still, it cannot establish a causal relationship between them.

Response: We appreciate the thoughtful consideration of causal relationships in the general discussion. The manuscript was revised to explicitly acknowledge the limitations in establishing causal relationships and address the potential confounding factors and reverse causality.

---

## [Decision Letter · Decision Letter 1]

30 Jan 2024

Relationship between gender roles, motherhood beliefs and mental health

PONE-D-23-22025R1

Dear Dr. Reyes-Aguilar,

We are pleased to inform you that your manuscript has been judged scientifically suitable for publication and will be formally accepted for publication once it meets all outstanding technical requirements

Within one week, you will receive an e-mail detailing the required amendments. When these have been addressed, you’ll receive a formal acceptance letter and your manuscript will be scheduled for publication.

Kind regards,

Sergi Fàbregues

Academic Editor

PLOS ONE

Additional Editor Comments:

Reviewer 1 has pointed out a few minor comments that the authors should take into account in the final version of the manuscript.

Reviewers' comments:

Reviewer's Responses to Questions

**Comments to the Author**

1. If the authors have adequately addressed your comments raised in a previous round of review and you feel that this manuscript is now acceptable for publication, you may indicate that here to bypass the “Comments to the Author” section, enter your conflict of interest statement in the “Confidential to Editor” section, and submit your "Accept" recommendation.

Reviewer #1: (No Response)

Reviewer #2: All comments have been addressed

2. Is the manuscript technically sound, and do the data support the conclusions?

Reviewer #1: Yes

Reviewer #2: Yes

3. Has the statistical analysis been performed appropriately and rigorously? 

Reviewer #1: Yes

Reviewer #2: Yes

4. Have the authors made all data underlying the findings in their manuscript fully available?

Reviewer #1: No

Reviewer #2: Yes

5. Is the manuscript presented in an intelligible fashion and written in standard English?

Reviewer #1: Yes

Reviewer #2: Yes

6. Review Comments to the Author

Reviewer #1: The authors have responded thoroughly and graciously to all my previous concerns regarding the manuscript. My remaining comments are minor.

1) It may be helpful for potential readers if the title includes that the study was conducted in Mexico.

2) The term “breeding” could be replaced with an academic term applicable to humans in particular (e.g., “procreation”, “reproduction”, or “bearing children”).

3) The authors have now included information on employment, but I would not describe it as data on “occupation” per se. The data reported Table 3 for Study 3 is more appropriately referred to as “employment status.” On the other hand, occupational data would, for instance, distinguish between professional or “white collar” occupations and “blue collar” occupations.

4) After thoughtful incorporation of reviewers’ feedback, with revisions, the paper is now 70 pages long. The detailed attention to the reviewer feedback is laudable. However, I might suggest moving some tables to an online Appendix section if the paper’s length becomes an issue.

Thank you again for the opportunity to review this important work.

Reviewer #2: The article is a contribution to the literature on gender and mental health. The authors are thanked for their contributions to the problematization of gender normativity in aspects such as people's mental health and in validating scales to measure it.

7. PLOS authors have the option to publish the peer review history of their article (what does this mean?). If published, this will include your full peer review and any attached files.

Reviewer #1: No

Reviewer #2: No

---

## [Editor Report · Acceptance letter]

22 Feb 2024

PONE-D-23-22025R1 

PLOS ONE

Dear Dr. Reyes-Aguilar, 

I'm pleased to inform you that your manuscript has been deemed suitable for publication in PLOS ONE. Congratulations! Your manuscript is now being handed over to our production team.

Kind regards, 

on behalf of

Dr. Sergi Fàbregues 

Academic Editor

PLOS ONE